# Enhanced daytime secondary aerosol formation driven by gas-particle partitioning in downwind urban plumes

Mingfu Cai[1,2,3], Ye Chenshuo[4], Bin Yuan[2,3*], Shan Huang[2,3], E Zheng[2,3], Suxia Yang[5], Zelong Wang[2,3], Yi Lin[2,3], Tiange Li [2,3], Weiwei Hu[6], Wei Chen[6], Qicong Song[2,3], Wei Li[2,3], Yuwen Peng[2,3], Baoling Liang[7], Qibin Sun[7], Jun Zhao[7], Duohong Chen[8], Jiaren Sun[1], Zhiyong Yang[9], Min Shao[2,3]

[1]Guangdong Province Engineering Laboratory for Air Pollution Control, Guangdong Provincial Key Laboratory of Water and Air Pollution Control, South China Institute of Environmental Sciences, MEE, Guangzhou 510655, China

[2]Institute for Environmental and Climate Research, Jinan University, Guangzhou 51143, China

[3]Guangdong-Hongkong-Macau Joint Laboratory of Collaborative Innovation for Environmental Quality, Jinan University, Guangzhou 510632, China

[4] Guangdong Provincial Academy of Environmental Science, Guangzhou, 510045, China

[5] Guangzhou Research Institute of Environment Protection Co.,Ltd, Guangzhou 510620, China

[6]State Key Laboratory of Organic Geochemistry and Guangdong Key Laboratory of Environmental Protection and Resources Utilization, Guangzhou Institute of Geochemistry, Chinese Academy of Sciences, Guangzhou 510640, China

[7]School of Atmospheric Sciences, Guangdong Province Key Laboratory for Climate Change and Natural Disaster Studies, and Institute of Earth Climate and Environment System, Sun Yat-sen University, Zhuhai 519082, China

[8]Guangdong Environmental Monitoring Center, Guangzhou 510308, China

[9]Guangzhou Huangpu District Meteorological Bureau, Guangzhou 510530, China

*Corresponding authors*: Bin Yuan (byuan@jnu.edu.cn)

**Abstract.**

Anthropogenic emissions from city clusters can significantly enhance secondary organic aerosol (SOA) formation in the downwind regions, while the mechanism is poorly understood. To investigate the effect of pollutants within urban plumes on organic aerosol (OA) evolution, a field campaign was conducted at a downwind site of the Pearl River Delta region of China in the fall of 2019. A time-of-flight chemical ionization mass spectrometer coupled with a Filter Inlet for Gases and Aerosol (FIGAERO-CIMS) was used to probe the gas- and particle-phase molecular composition and thermograms of organic compounds. For air masses influenced by urban pollution, strong daytime SOA formation through gas-particle partitioning was observed, resulting in higher OA volatility. The obvious SOA enhancement was mainly attributed to the gas-particle partitioning of high volatility (SVOC+IVOC+VOC, $C^* > 0.3$ μg m$^{-3}$) organic vapors. Using the equilibrium equation could underestimate the contribution of high volatility organic vapors, since the volatility of these species in the particle-phase was lower than that in the gas-phase. We speculated that the elevated NO$_x$ concentration could suppress the formation of highly oxidized products, resulting in a smooth increase of low volatility (ELVOC+LVOC, $C^* \leq 0.3$ μg m$^{-3}$) organic vapors. Evidences showed that urban pollutants (NO$_x$ and VOCs) could enhance the oxidizing capacity, while the elevated VOCs were mainly responsible for promoting daytime SOA formation by increasing the RO$_2$ production rate. Our results highlight the important role of urban anthropogenic pollutants in SOA control in the suburban region.

## 1. Introduction

As a major concern of air pollution, aerosol particles are known to have significant impacts on public health and climate (Apte et al., 2018; Arias et al., In Press). Primary particulate matter (PM) in China has shown a remarkable reduction since 2013, owing to strictly clean air policies implemented by the Chinese government (Zhang et al., 2019). Despite the effective reduction of primary emissions in the past ten years, secondary organic aerosol (SOA) remains at high levels and is mainly responsible for the haze development in China(Huang et al., 2014). SOA is thought to be formed through the oxidation of volatile organic compounds (VOCs) and atmospheric aging processes of primary organic aerosol (POA). However, models are especially challenged in reproducing SOA concentration and properties, since the formation mechanisms and gas precursors of SOA remain poorly characterized(Hodzic et al., 2010).

Gas-particle partitioning of organic vapors is found to be the important formation pathway of SOA the worldwide(Nie et al., 2022; Hallquist et al., 2009; Lanzafame et al., 2021). Nie et al. (2022) suggested that the contribution of the condensation of organic vapors to the SOA mass growth ranged from about 38%-71% in China megacities. Photochemical produced SOA via gas phase chemistry is usually related to a higher volatility and a lower oxidation degree than that formed in the aqueous phase (Ervens et al., 2011; Saha et al., 2017). The condensation processes of organic vapors are determined by their volatility, which is closely related to oxidation state, functional groups, and the number of atomic carbons. Laboratory studies revealed that high nitrogen oxides ($NO_x$) concentration can suppress the production of molecules with high oxidation degree by inhibiting autoxidation(Rissanen, 2018; Peng et al., 2019), which is considered to be an important pathway of low volatility vapor formation(Praske et al., 2018). Such compounds have been shown to play a vital role in the SOA formation and growth of newly formed particles(Mutzel et al., 2015; Bianchi et al., 2019; Mohr et al., 2019). On the other hand, it is shown that the increase of oxidant owing to elevated $NO_x$ concentration can offset the decrease of autoxidation efficiency, leading to a higher production of oxygenated organic vapors(Pye et al., 2019), highlighting the complexity of SOA formation. However, the lack of a molecular dataset of SOA and gas precursors hinders the understanding of the SOA formation mechanism.

Recently, a chemical ionization time-of-flight mass spectrometer coupled with a Filter Inter for

Gases and AEROsols (FIGAERO-CIMS) has been employed to measure gas- and particle-phase
oxygenated organic compounds the worldwide (Chen et al., 2020; Buchholz et al., 2020; Masoud et
al., 2022). Using a FIGAERO-CIMS, Cai et al. (2023) showed that heterogeneous reaction might
have an important role in the secondary formation of particle-phase oxidized organic nitrogen. The
volatility of OA can provide information about the formation and aging processes of OA, given that
it is strongly affected by chemical composition. In past decades, a thermodenuder (TD) coupled
with aerosol detection instruments (e.g. aerosol mass spectrometer and condensation particle
counter) was widely used in the estimation of OA volatility (Philippin et al., 2004; Lee et al., 2010).
Cai et al. (2022) found that the OA volatility was higher at a particle size range of 30 to 200 nm
during daytime, suggesting that the SOA formation through gas-particle partitioning could generally
occur at all particle sizes. However, this method failed to provide the volatility information of
different molecules of OA. In recent years, the FIGAERO-CIMS was developed to characterize the
volatility of oxygenated organic molecules in the particle phase. (Ren et al., 2022; Ylisirniö et al.,
2020). Wang and Hildebrandt Ruiz (2018) showed that the thermal desorption products of SOA can
be separated into different groups on a two-dimensional thermogram measured by the FIGAERO-
CIMS. Ren et al. (2022) investigated the relationship between the molecular formulae of OA
components and their volatilities and suggested that the volatility of OA compounds was strongly
affected by O to C ratio. These results provide valuable insights into the SOA formation mechanisms.
However, as yet few FIAGERO-CIMS field studies are available in the literature in China(Ye et al.,
2021; Salvador et al., 2021), especially in urban downwind areas.

Observational studies have demonstrated that anthropogenic emissions can significantly affect

SOA formation in the downwind region. Fry et al. (2018) observed an enhancement of organic
nitrate aerosol formed through $NO_3$+isoprene in power plant plume during nighttime, which was
mainly attributed to $NO_x$ emissions from the power plant. The results from Liu et al. (2018)
suggested that the OH concentrations increased by at least 250% under polluted conditions, which
might promote the daytime SOA formation. A field measurement in the Amazon forest by De Sá et
al. (2018) showed that the enhancement of OA (about 30-171%) in urban plumes was mainly
contributed by SOA.  A recent study founded that anthropogenic emission of $NO_x$ from urban could
enhance oxidant concentration, thereby promoting daytime SOA formation(Shrivastava et al., 2019).

In this study, we investigate the SOA formation through photochemical reactions at a typical

downwind site in the Pearl River Delta region (PRD) using a FIGAERO-CIMS along with a suite of other online instruments. The volatility of OA and its relationship with identified OA sources during long-range transport, urban air masses, and coastal air masses periods are discussed. The formation mechanisms of daytime SOA formation within the urban plume are investigated based on online measurements of gas- and particle-phase organic compounds, gaseous pollutants, and aerosol physicochemical properties. The impact of urban pollutants on SOA formation will be discussed.

## 2. Measurement and Method

### 2.1 Field measurement

The campaign was conducted at the Heshan supersite in the PRD region during the fall of 2019 (29[th] September to 17[th] November 2019). The Heshan Supersite, surrounded by farms and villages, is located (at 22°42′39. 1″N, 112°55′35.9″E, with an altitude of about 40 m) at southwest of the PRD region and about 70 km southwest of Guangzhou city (Fig. S1). During the measurement, the sampling site is mainly influenced by the air masses from the center of the PRD region (Fig. S2a). All instruments were placed in an air-conditioned room on the top floor of the supersite. A detailed description of the site and experimental setup can be found in Cai et al. (2021).

### 2.2 Instrumentation

### 2.2.1 FIGAERO-CIMS

A FIGAERO-CIMS coupled with an X-ray source was employed to measure organic compounds in the gas- and particle-phase using I$^-$ as the chemical ionization reagent. The particle sampling inlet of the FIGAERO-CIMS was equipped with a PM$_{2.5}$ cyclone and a Nafion dryer (model PD-07018T-12MSS, Perma Pure, Inc., USA). The principle of the instrument can be found in Lopez-Hilfiker et al. (2014) and Le Breton et al. (2018). In general, the operation settings and data processing were the same as Cai et al. (2023) and Ye et al. (2021). Here, only a brief description relevant to the measurement is given. The instrument was worked in a cycle pattern of 1 hour, with 24 minutes of gas-phase measurements and particle collection (sampling mode), followed by a 36-minutes particle-phase analysis (desorption mode). In the sampling mode, ambient gas was

measured in the first 21 minutes, followed by a 3-min zero air background. At the same time,
ambient particles were collected on a PTFE membrane filter. In the desorption mode, the collected
particles were desorbed by heated $N_2$. The temperature of the $N_2$ was linearly ramped from indoor
temperature (~25°C) to ~175 °C in 12 minutes and held for 24 minutes. The data processing steps
in this campaign were the same as Ye et al. (2021). A few chemicals were calibrated before and after
the measurement. For uncalibrated species, a voltage scanning method was employed to obtain their
sensitivities (referred to as semi-quantified species) (Ye et al., 2021; Iyer et al., 2016; Lopez-Hilfiker
et al., 2016).
**2.2.2 SP-AMS**
The $PM_1$ chemical composition was measured by a soot particle aerosol mass spectrometer
(SP-AMS, Aerodyne Research, Inc., USA). The details of the operation and data analysis can be
found in Kuang et al. (2021). Source apportionment was performed for organic aerosols in the bulk
$PM_1$ using positive matrix factorization (PMF). The organic aerosol could be divided into six
components, including two primary OA factors and four secondary OA factors. The mass spectral
profiles of six OA factors are shown in Figure S3. The timeseries and diurnal variation of these
factors are presented in Figure S4.
The primary OA factors include hydrocarbon-like OA (HOA), mainly contributed by traffic
and cooking emissions and biomass burning OA (BBOA) originating from biomass burning
combustion. The HOA was identified by hydrocarbon ions $C_xH_y^+$. Owing to the prominent
hydrocarbon ions and low O:C value (0.10), HOA could be attributed to primary emission from
cooking and traffic. The BBOA was recognized by the markers $C_2H_4O_2^+$ (m/z 60.022, 0.5%) and
$C_3H_5O_2^+$ (m/z 73.029, 0.4%), which are considered tracers for biomass burning OA (Ng et al., 2011).
The SOA factors include biomass burning SOA (BBSOA) likely formed from oxidation of
biomass burning emission, less oxygenated OA (LOOA) provided by strong daytime photochemical
formation, more oxygenated OA (MOOA) related to regional transport, and nighttime-formed OA
(Night-OA) contributed by secondary formation during nighttime. The BBSOA was likely formed
through oxidation of biomass burning precursors, which was supported by the evening peak at about
19:00 LT (Fig. S4). BBSOA showed a similar variation trend with $C_6H_2NO_4^+$, which might be
contributed by oxidation of gaseous precursors from biomass burning emissions (Wang et al., 2019;
Bertrand et al., 2018). The significant afternoon peak of LOOA indicates its formation through
photochemical reactions, which would be detailly discussed in section 3.1. The negligible diurnal
variation and the highest O:C value (1.0) of MOOA suggested that it could be aged OA resulting
from long-range transport. Night-OA was formed through $NO_3$ nighttime chemistry, supported by
a pronounced evening elevation and positive correlation with nitrate (R=0.67).The detailed
determination of PMF factors has been found in Kuang et al. (2021) and Luo et al. (2022).
**2.2.3 Particle number size distribution measurements**
Particle number size distribution in a size range of 1 nm - 10 μm was measured by a diethylene
glycol scanning mobility particle sizer (DEG-SMPS, model 3938E77, TSI Inc., USA), a SMPS
(model 3938L75, TSI Inc., USA), and an aerodynamic particle sizer (APS, model 3321, TSI Inc.,
USA). All sample particles first passed through a Nafion dryer (Model MD-700, Perma Pure Inc.,
USA) to reduce relative humidity (RH) lower than 30%. A detailed description of these instruments
can be found in Cai et al. (2021).
**2.2.4 Other parameters**
The non-methane hydrocarbons(NMHC)were measured by an online GC-MS-FID (Wuhan
Tianhong Co., Ltd, China). The concentration of oxygenated VOCs, including formaldehyde
(HCHO) and acetaldehyde ($CH_3CHO$), were measured using high-resolution proton transfer
reaction time-of-flight mass spectrometry (PTR-ToF-MS, Ionicon Analytik, Austria). HONO was
detected by the gas and aerosol collector (GAC) instrument (Dong et al., 2012). Trace gases,
including $O_3$, $NO_x$, and CO, were measured by gas analyzers (model 49i, 42i, and 48i, Thermo
Scientific, US). Meteorological parameters (i.e., wind speed, wind direction, and temperature) were
measured by a weather station (Vantage Pro 2, Davis Instruments Co., US).
**2.3 Methodology**
**2.3.1 Estimation of the volatility of particle- and gas-phase organic compounds**
During the heating processes, the FIGAERO-CIMS simultaneously measured the desorbing
compounds of the collected particles. Thus, the volatility information of particles can be obtained
by investigating the relationship between the measured signals and desorption temperature. The
temperature of the peak desorption signal ($T_{max}$) has a nearly linear relationship with the natural
logarithm of saturation vapor pressure ($P_{sat}$) of the respective compound (Lopez-Hilfiker et al.,

2014):

$$ln(P_{sat}) = aT_{max} + b \qquad (1)$$
where $a$ and $b$ are fitting coefficients. Thus, saturation vapor concentration ($C^*$, μg m$^{-3}$) can be
obtained:
$$C^* = \frac{P_{sat}M_w}{RT}10^6 \qquad (2)$$
where $M_w$ is the molecular weight of the compound (determined by the FIGAERO-CIMS), R is the
universal gas constant (8.314 J mol$^{-1}$ K$^{-1}$), and T is the thermodynamic temperature in kelvin (298.15
K).
We used a series of polyethylene glycol (PEG 5-8) compounds to calibrate the $T_{max}$ and
obtained the fitting parameters $a$ and $b$. The PEG standards were prepared in a mixture of
acetonitrile and then atomized with a homemade atomizer. The atomized particles are classified by
a differential mobility analyzer (DMA, model 3081 L, TSI Inc., USA) at two diameters (100 nm
and 200 nm). The selected particles were then split into two paths: one to a condensation particle
counter (CPC, model 3775, TSI Inc., USA) for measuring the particle concentration and another
one to the particle inlet of the FIGAERO-CIMS. The collected concentration can be calculated based
on the selected particle diameter, particle number concentration, flow rate of the particle inlet of
FIGAERO-CIMS, and collection time. The calibration results and corresponding fitting parameters
can be found in Fig. S5 and Table. S1. Note that the $T_{max}$ can vary with mass loading, and it is
necessary to consider for estimation the relationship between $T_{max}$ and $C^*$ (Wang and Hildebrandt
Ruiz, 2018). Our calibration results demonstrated that the correlation between $T_{max}$ shift and mass
loading was not linear, which may be attributed to matrix or saturation effects (Huang et al., 2018).
During the measurement, the collected mass loading centered at about 620 ng and the particle
volume size distribution (PVSD) centered at about 400 nm (Fig. S6). Thus, the fitting parameters
(a=-0.206 and a=3.732) of the calibration experiment with a diameter of 200 nm and mass loading
of 407 ng were adopted in the $C^*$ calculation, since the mass loading and diameter are the closest to
the ambient samples.
For gas-phase organic compounds (organic vapors), we first divided them into two groups
based on their oxidation pathways (multi-generation OH oxidation and autoxidation, solid line in
Fig. S7) and then used different parameters in their volatility estimation. The classification of
pathways was based on the molecular characteristics of oxidation products of aromatics and
monoterpene, respectively (Wang et al., 2020). In general, their saturation vapor concentration ($C^*$,
at 300 K) can be estimated as follows:
$$log_{10}(C^*(300K)) = (25 - n_c) \cdot b_C - (n_O - 3n_N) \cdot b_O - \frac{2(n_O - 3n_N)n_C}{(n_C + n_O - 3n_N)} \cdot b_{CO} - n_N \cdot b_N \quad (3)$$
where $n_c$, $n_O$, and $n_N$ are the numbers of carbon, oxygen, and nitrogen atoms in each compound.
For oxidation products formed from multi-generation OH oxidation (aging) pathway, the volatility
parameters $b_C$, $b_O$, $b_{CO}$, and $b_N$ were assumed to be 0.475, 2.3, -0.3, and 2.5, respectively (Donahue
et al., 2011). For oxidation products formed from autoxidation pathway, the modified
parameterization is used, with $b_C$=0.475, $b_O$=0.2, $b_{CO}$=0.9, and $b_N$ =2.5 (Bianchi et al., 2019). It
should be noted that this method can only roughly distinguish the formation pathways of ambient
organic vapors, since it is based on the oxidation products of specific species in a laboratory study.

### 2.3.2 Calculation of oxidation state ($\overline{OS_C}$) of $C_xH_yO_z$ and $C_xH_yN_{1,2}O_z$ compounds

For $C_xH_yO_z$ compounds, the $\overline{OS_C}$ can be estimated as:
$$\overline{OS_C} = 2 \times \frac{O}{C} - \frac{H}{C} \quad (4)$$
For $C_xH_yN_{1,2}O_z$ compounds, the $\overline{OS_C}$ can be calculated from following equation:
$$\overline{OS_C} = 2 \times \frac{O}{C} - \frac{H}{C} - x \times \frac{N}{C} \quad (5)$$
where $x$ is the valence state of N atoms, which is dependent on functional groups. Several
assumptions were adopted to classify them. (1)N-containing functional groups were nitro (-NO$_2$,
$x$=+3) or nitrate (-NO$_3$, $x$=+5) in our measurement; (2)N-containing aromatics contain nitro
moieties while N-containing aliphatic hydrocarbons contain nitrate moieties; (3)N-containing
aromatics have 6-9 carbon atoms and fewer hydrogen atoms than aliphatic hydrocarbons with the
same number of carbon atoms.

### 2.3.3 Estimation of condensation sink

The condensation sink (CS) represents the condensing vapor captured by pre-existing particles and can be calculated from the following equation:

$$CS = 2\pi D \sum_{D_p} \beta_{m,D_p} D_p N_{D_p} \tag{6}$$

where $D$ is the diffusion coefficient of the $H_2SO_4$ vapor ($0.8 \times 10^{-5}$ m$^2$ s$^{-1}$), $\beta_{m,D_p}$ is the transitional regime correction factor which can be calculated from the Knudsen number (Fuchs and Sutugin, 1971), and $N_{D_p}$ represents the particle number concentration at $D_p$.

### 2.3.4 Estimation of OA contributed by high volatility organic vapors

Organic vapors with higher volatility (SVOC+IVOC+VOC, $C^* > 0.3$ µg m$^{-3}$) can easily reach an equilibrium between the gas and particle phase. Thus, the contribution of high volatility organic vapors to OA concentration ($OA_{HVgas}$) through gas-particle partitioning can be estimated as following:

$$OA_{HVgas} = \sum_i C_{i,g} f_i \tag{7}$$

where $C_{i,g}$ is the gas-phase concentration of species $i$. $f_i$ is the fraction of species $i$ in the particle phase and is defined as:

$$f_i = \frac{C_{OA}}{C_{OA} + C_i^*(T)} \tag{8}$$

where $C_{OA}$ is the concentration of OA measured by the SP-AMS, and $C_i^*(T)$ is the saturation concentration of species $i$ at temperature ($T$). The temperature-dependent $C_i^*(T)$ was obtained by (Nie et al., 2022):

$$\log_{10} C_i^*(T) = \log_{10} C_i^*(300K) + \frac{\Delta H_{vap,i}}{R \ln(10)}\left(\frac{1}{300} - \frac{1}{T}\right) \tag{9}$$

$$\Delta H_{vap,i} = -5.7 \log_{10} C_i^*(300K) + 129 \tag{10}$$

where $\Delta H_{vap,i}$ is the enthalpy of vaporization and can be estimated based on $\log_{10} C_i^*(300K)$.

### 2.3.4 Estimation of the production rate of RO$_2$ and OH

A zero-dimensional box model (0-D Atmospheric Modeling, F0AM(Wolfe et al., 2016)) based on Master Chemical Mechanism (MCM v3.1.1, https://mcm.york.ac.uk/MCM) was used to simulate the production rate of OH in this study. The F0AM box model has been widely used in investigating

chemical reactions of VOCs, NO$x$, and RO$_x$ radicals (including OH, HO$_2$, and RO$_2$) in field and
laboratory researches (Baublitz et al., 2023; Yang et al., 2022; D'ambro et al., 2017). The simulation
was constrained with the observation data of non-methane hydrocarbons (NMHC), HCHO,
CH$_3$CHO, NO, CO, HONO, and meteorological parameters (RH, temperature, photolysis rates, and
pressure). The background concentration of CH$_4$ was set as 1.8 ppm (Wang et al., 2011). The
simulation time step was set to be 5 minutes. With respect to the integrity and temporal coverage of
the observation data, the simulation period was from 16 October to 16 November 2019. Further
details on model settings can be found in Yang et al. (2022)
The empirical kinetic modeling approach (EKMA) is applied to investigate the sensitivity of
the production rate of RO$_2$ and OH to the variation of NO$_x$ and VOCs. The base case was simulated
based on the observation of average conditions. Sensitivity tests are performed by adjusting NO$_x$ or
VOCs by a ratio ranging from 0.1 to 2.0 without changing other parameters.

## 3. Results and discussion

**3.1 Overview**

Figure 1 shows the temporal profile of particle number size distribution (PNSD) and
condensation sink (CS) during the measurement (a), one-dimensional thermograms and $T_{max}$
measured by the FIGAERO-CIMS (b), bulk PM$_1$ chemical composition measured by the SP-AMS
and PM$_1$ concentration (c), deconvolved OA factors from PMF analysis (d), and wind speed and
direction (e). Note that all measurements started on 2 October. As shown in Fig. 1a, new particle
formation (NPF) events occurred frequently along with relatively low CS values during the
measurement period (44.4%, 20 out of 45 days). The $T_{max}$ mainly varied in two temperature ranges,
80-95 °C and 110-120°C (Fig. 1b). The lower $T_{max}$ was usually accompanied by high desorption
signals peaked at 80-95 °C (Fig. 1b), a higher fraction of LOOA (Fig. 1d), and an obvious wide
accumulation mode in PNSD (Fig. 1a).
The evening peak of hydrocarbon-like OA (HOA) and biomass burning OA (BBOA) was
related to local anthropogenic activities (e.g., biomass burning, cooking, and traffic, Fig. 2). The
less oxygenated OA (LOOA) and biomass burning SOA (BBSOA) showed afternoon peaks (Fig.

2), which could be attributed to secondary organic aerosol (SOA) formation through daytime photochemical reactions. LOOA showed a noticeable increase corresponding to the particle surface area (Fig. S8), while we did not observe such correlation for other SOA factors (MOOA and BBSOA). Furthermore, LOOA exhibited a stronger positive correlation with organic vapors measured by the FIGAERO-CIMS compared to other OA factors (Fig. S9). These results suggested that the daytime formation of LOOA was attributed to gas-particle partitioning. The $O_x$ ($O_x = O_3 + NO_2$) had a strong correlation with organic vapors in the afternoon (10:00-16:00 LT, Fig. S10), highlighting an important role of photochemical reaction on the formation of LOOA.

The high desorption signal at a lower temperature range suggested that the volatility of OA could be higher, which could be associated with the formation of LOOA. Coincidently, either NPF events or a higher fraction of LOOA could only be observed during the period prevalent with north wind direction (Fig. 1e), when the measurement site was affected by the pollutant from the city cluster around Guangzhou city. It indicates that the urban pollutants might promote particle formation and growth and daytime SOA formation by increasing oxidants and acting as precursor gases. Xiao et al. (2023) suggested that fresh urban emissions could enhance NPF, while NPF was suppressed in aged urban plumes. Shrivastava et al. (2019) found that urban emissions, including $NO_x$ and oxidants, could significantly enhance the SOA formation in the Amazon rainforest. Three periods were classified based on the combination of wind direction and the analysis of backward trajectories to further investigate the impact of urban pollutants on this downwind site, which were long-range transport, urban air masses, and coastal air masses periods (Fig. S2 and Table. S2). The long-range transport period was related to long-range transport masses from northeast inland. The urban air masses period was mainly affected by regional urban air masses from the PRD region. The coastal air masses period was associated with air masses from the South China Sea and the northeast coast.

A significant daytime peak of LOOA (10.4 $\mu g\ m^{-3}$) was shown during the urban air masses period (Fig. 2c), while the enhancement of BBSOA was inapparent. It suggests that the contribution of gas-particle reactions on SOA formation was enhanced when the site was affected by urban plumes. The $O_x$ concentration in the afternoon during the urban air masses period was higher than that during the long-range transport period (Fig. S11), which might be able to explain the significant enhancement of LOOA for the urban air masses period. These results imply that urban pollution

plumes could promote the formation of SOA in the downwind region by increasing the oxidant
concentration.
**3.2   The daytime formation of FIGAERO OA**

As aforementioned, the increase of LOOA was usually along with the significant desorption

signals measured by the FIGAERO-CIMS at a low temperature range (80-95°C), suggesting that
OA volatility could be higher. The average two-dimensional thermograms of all calibrated and semi-
quantified species and an example of a one-dimensional thermogram of levoglucosan can be found
in Fig. 3 a and b, respectively. According to Eqs. (1) and (2), we calculated the $C^*$ value of all
calibrated and semi-quantified species based on their $T_{max}$ and constructed volatility distribution as
volatility basis set (VBS, Fig. 3c). The $T_{max}$ of each species is obtained based on their average
thermogram. These 12 VBS bins were classified into three groups(Donahue et al., 2012): semi-
volatile organic compounds (SVOC, $0.3 < C^* \leq 3 \times 10^2$ μg m$^{-3}$), less-volatile organic compounds
(LVOC, $3 \times 10^{-4} < C^* \leq 0.3$ μg m$^{-3}$), and extremely low-volatility organic compounds (ELVOC,
$C^* \leq 3 \times 10^{-4}$ μg m$^{-3}$). In general, most species measured by FIGAERO-CIMS fall into LVOC groups
(Fig. S12). Note that the decomposition of organic compounds was ignored in this method, which
could affect thermogram peaks in some cases and the measurement of low volatility compounds
(Wang and Hildebrandt Ruiz, 2018). Furthermore, the fraction of SVOC might be underestimated
owing to its high volatility, as a result fast evaporation could occur during the collection on the filter
and shifting from sampling mode to desorption mode.

During the urban air masses period, the FIGAERO-CIMS measured significant signals at a

desorption temperature range of SVOC and LVOC (Fig. S13) in the afternoon (12:00-16:00 LT),
indicating that the OA volatility could be higher. The SVOC+LVOC in the FIGAERO OA increased
from 5.2 μg m$^{-3}$ (8:00 LT) to 16.29 μg m$^{-3}$ (15:00 LT) during the urban air masses period (Fig. 4a),
which was coincident with an enhancement of LOOA (Fig. 2c). It suggested that daytime
enhancement of the SVOC+LVOC in the FIGAERO OA was closely related to the obvious LOOA
formation. The FIGAERO OA during the urban air masses period was systemically higher than that
during the long-range transport period, with a significantly higher concentration of LVOC group
(Fig. 4b), especially the portion with a volatility $log_{10}C^*$ of -1. Table 1 investigated the relationship
between SVOC+LVOC and six OA factors. The SVOC+LVOC in FIGAERO OA had a significant
positive correlation (R=0.72-0.85) with the LOOA, especially during the urban air masses period
(R=0.85, Fig. S14 and Table 1), suggesting that the LOOA formation was mainly responsible for
the increase of OA volatility.

Interestingly, the high volatility organic vapors (SVOC+IVOC+VOC, $C^* > 0.3$ μg m$^{-3}$)

dramatically increased in the afternoon during the urban air masses period, while we did not observe
such phenomenon for low volatility (ELVOC+LVOC, $C^* \leq 0.3$ μg m$^{-3}$) organic vapors (Fig. 4c).
The concentration of low volatility organic vapors in the afternoon (12:00-16:00 LT) did not show
a significant difference (1.76 and 1.84 μg m$^{-3}$) between the long-range transport and urban air masses
periods, indicating that the irreversible condensation of low volatility organic vapors could not fully
explain the enhancement of LOOA during the urban air masses period (Wang et al., 2022). However,
the high volatility organic vapors had a notably higher concentration (51.69 μg m$^{-3}$) during the urban
air masses period than that (41.70 μg m$^{-3}$) during the long-range transport period. It implies that the
significant enhancement of LOOA during the urban air masses period might be mainly attributed to
the equilibrium partitioning of high volatility organic vapors, which could also increase the volatility
of total OA.

Here we selected a typical day (2 November 2019) of the urban air masses period for further

investigation. The measurement site was affected by the urban plume from the city cluster in the
PRD region on this day (Fig. S15). A wide accumulation mode centered at about 180 nm in PNSD
was observed, with a significant desorption signal measured by the FIGAERO-CIMS in the
afternoon and weak north wind (Fig. S16). As shown in Fig. 5a, the desorption signals of organic
compounds increased from 9:00 LT and reached their peak at 14:00 LT, suggesting a significant
daytime SOA formation. The variation of OA volatility distribution and mean $C^*(\overline{C^*})$ is shown in
Fig. 5b. The $\overline{C^*}$ shown an afternoon peak (0.021 μg m$^{-3}$) at 15:00 LT, suggesting a higher OA
volatility in the afternoon. An evident enhancement of OA with a volatility $log_{10}C^*$ of -1 was
observed in the afternoon, aligning with the formation of LOOA (Fig. 5c), which primarily
contributes to higher OA volatility. Combined with the volatility distribution analysis in Fig. 4b, it
indicated that the main components of LOOA have a volatility $log_{10}C^*$ of -1. Interestingly, the
$T_{max}$ value of the sum thermogram (Fig. 5a) increased from 81°C at 9:00 to 96°C at 17:00, implying
that the OA volatility decreased during the daytime owing to the daytime aging processes. However,
the $\overline{C^*}$ value consistently increased from 6:00 LT until 15:00 LT and then began to decrease, which
was conflict with the increasing $T_{max}$. One possible reason is that species in the FIGAERO OA fell
into a specific $T_{max}$ range (about 11°C) were categorized into different $C^*$ bins by a factor of 10.
Thus, the slight variation of $T_{max}$ might not affect the estimated volatility distribution of FIGAERO
OA. The other possible reason is that the volatility distribution of FIGAERO OA was estimated
based on the $T_{max}$ value of calibrated and semi-quantified species, while the sum thermograms
contained all organic compounds containing C, H, and O atmos. There could be some organic
compounds formed through aging processes that were not included in the $C^*$ estimation.
**3.3 The contribution of high volatility organic vapors to SOA formations**
In the previous section, we found that the significant enhancements in LOOA during the urban
air masses period might be attributed to the high volatility organic vapors through gas-particle
partitioning. The contribution of high volatility organic vapors to the OA concentration via
equilibrium partitioning can be estimated based on eq. (7). Our results show that the estimated
contribution of high volatility organic vapors (estimated OA$_{HVgas}$) was higher (peaked at about 1.17
μg m$^{-3}$) during the urban air masses period (Fig. 6a). Correspondingly, we observed an enhancement
in the measured concentration of these species in the particle-phase (measured OA$_{HVgas}$, peaked at
about 10.32 μg m$^{-3}$, Fig. 6b). This implies that the increase in high volatility organic vapors might
significantly contribute to the daytime SOA formation during the urban air masses period. However,
the estimated contribution was much lower than the measured value. It suggests that using the
equilibrium equation might not be able to fully explain the increase of LOOA contributed by the
high volatility organic vapors during the urban air masses period. Nie et al. (2022) indicated that the
estimation of OA contribution through the equilibrium equation can be easily disturbed by varied
meteorological processes, which would lead to uncertainties in the calculations.
Moreover, the gas-particle equilibrium theory assumes that particles are droplets and that the
high volatility species in the particle-phase could reach a reversible equilibrium with the gas-phase
concentration. However, some studies indicate that this assumption significantly overestimates the
volatility of these species in the particle-phase and underestimate the contribution of high volatility
organic vapors to the SOA concentration (Kolesar et al., 2015; Cappa and Wilson, 2011). This is
because particles might exist in a glassy state rather than a liquid state. It was consistent with the
difference of the volatility distribution of these species between the particle- and gas-phase (Fig.
7a). The volatility in the particle-phase was centered at a $log_{10}C^*$ of -1, while that in the gas-phase
showed a higher concentration of $log_{10}C^*$=6-8 µg m$^{-3}$, implying that the volatility of these
compounds in the particle-phase could lower than that in the gas-phase.

Another possible explanation is that the corresponding species in the particle-phase could be

the decomposition products of low volatility compounds, leading to a higher concentration than
expected. We further investigate the difference between the measured and estimated concentration
of different high volatility species (Fig. 7b). The measured concentration was systematically higher
than the estimated value. The higher measured concentration of $C_2H_2O_4I^-$ could be owing to the
decomposition of low volatility spices, as the desorption signal peaked at the ELVOC region (Fig.
7c). However, for higher molecular weight compounds, the corresponding $T_{max}$ values were in the
LVOC region, suggesting that these species might not be the decomposition products. This implies
that the decomposition products might play a minor effect in the difference between the measured
and estimated concentration.

Taken together, these results suggest the increase in high volatility organic vapors could

promote the daytime enhancement of SOA during urban air masses period. However, this
contribution might be underestimated using gas-particle equilibrium theory, since the volatility of
organic aerosol may differ significantly from the volatility determined by the equilibrium theory.
**3.4 Enhancement of SOA formation by urban pollutants**

As aforementioned, the significant enhancement of high volatility organic vapors were

observed during the urban air masses period. Figure 8 compares the difference of organic vapors in
the carbon oxidation state ($\overline{OS_C}$) in the afternoon (12:00-16:00 LT) between the long-range transport
and urban air masses periods. A higher concentration of organic vapors with a low $\overline{OS_C}$ ($\overline{OS_C} \leq 0$)
was observed during the urban air masses period, while this trend became to overturn for high $\overline{OS_C}$
($\overline{OS_C}$>0) organic vapors. It suggests that the oxidation degree of organic vapors was lower during
the urban air masses period, even though the $O_x$ concentration was higher (Fig. S11). This trend was
more significant for carbon number between 2 and 5, indicating a higher concentration of small
molecules with low $\overline{OS_C}$ during urban air masses period. The $\overline{OS_C}$ of major $C_5$ compounds was
about -1.33, which was mainly contributed by $C_5H_8NOI^-$, highlighting the role of $NO_x$ chemistry.
The oxygenated organic vapors production rates depend on oxidant and precursor concentration,
and the mechanism of significant enhancement of high volatility organic vapors remain unclear. We
speculated that it could be partly attributed to the elevated $NO_x$ concentration in the afternoon during
the urban air masses period (Fig. S17). $NO_x$ was found to have a detrimental effect on the production
of highly oxidized products, and thus the formation of low volatility vapors (Rissanen, 2018), which
might be responsible for the smooth increase of low volatility organic vapors. Previous studies found
that the increase of $NO_x$ could lead to higher OH production, which would offset decreases in the
autoxidation efficiency and further result in enhanced SOA formation (Liu et al., 2021; Pye et al.,
2019). During the urban air masses period, both low volatility and high volatility CHON compounds
increased in the afternoon, implying the effect of $NO_x$ on the photochemical reactions (Fig. S18 a
and b). That was further evidenced by the higher fraction of CHON compounds in the FIGAERO
OA (Fig. S18f). This result was consistent with Schwantes et al. (2019), who reported that low
volatility organic nitrates might have a significant contribution to SOA under high $NO_x$ conditions.
Interestingly, in contrast with the higher fraction of low volatility CHON compounds in the
afternoon, the fraction of high volatility CHON compounds was lower at the same time (Fig. S18 d
and e), indicating that the effect of high $NO_x$ concentration on photochemical oxidation goes beyond
the formation of CHON compounds for high volatility species.

To further understand how the urban plumes affect the SOA formation, we used an observation-

constrained box model to simulate the production rate of organic peroxy radicals ($RO_2$) and OH
with different $NO_x$ and VOCs concentrations (Fig. 9). The detailed description of the box model is
described in Sect. 2.3.4. In general, the production rates of OH (P(OH)) were close to the transition
regime during three selected periods (Fig. 9a), where the P(OH) is sensitive to both VOCs and $NO_x$
variation. Further, the P(OH) tended to be in the $NO_x$-limited regime during the coastal air masses
period. The emission of $NO_x$ might enhance the atmospheric oxidation capacity, consistent with the
results from other observations (Shrivastava et al., 2019; Pye et al., 2019). Interestingly, the
sensitivity regime of P(OH) changed to the VOCs-limited during the urban air masses period,
suggesting that the production of OH would be suppressed with continued increases in $NO_x$. During
the urban air masses period, the concentration of $NO_x$ and VOCs was noticeably increased compared
to the coastal air masses period, leading to a significant increase of P(OH).
Recent studies show that autooxidation of $RO_2$ can result in highly oxygenated molecules
(O:C≥0.7) and promote SOA formation(Pye et al., 2019; Pye et al., 2015). In general, the production
rate of $RO_2$ (P($RO_2$)) was in the VOCs-limited regime during three selected periods (Fig. 9b), where
the P($RO_2$) increased with the increase of VOCs. It suggests that the production of $RO_2$ was
suppressed with the increase in $NO_x$. During the urban air masses period, the concentration of VOCs
was noticeably increased compared to the coastal air masses period, leading to a significant increase
of P($RO_2$). The model results indicate that urban pollutants, including $NO_x$ and VOCs, could
enhance the oxidizing capacity, while the increase of VOCs was mainly responsible for significant
daytime SOA formation.

## 4. Conclusions

In this study, we demonstrated that daytime SOA formation could be enhanced when the rural
site was affected by the pollutant from the city region, which could be partly attributed to the high
concentration of oxidant in the urban pollution. A higher volatility of OA was observed during the
urban air masses period, which was mainly contributed by the component with a volatility $log_{10}C^*$
of -1. The significant increase of SVOC+LVOC in FIGAERO OA in the afternoon was associated
with enhanced LOOA formation. Similar to other measurements, the daytime formation of LOOA
was mainly through gas-to-particle partitioning of organic vapors, supported by a significant
positive relationship between the LOOA and organic vapors. We observed a dramatic increase in
the high volatility organic vapors in the afternoon during the urban air masses period, while low
volatility organic vapors did not exhibit a similar growth trend. It indicated that the rapid increase
of LOOA during the urban air masses period was mainly contributed by the gas-particle partitioning
of high volatility organic vapors. However, this contribution was underestimated using equilibrium
theory, since the volatility of high organic vapors in the particle phase was significantly lower than
that in the gas-phase.
The high $NO_x$ might also suppress the formation of highly oxidized products. Thus, the
elevated $NO_x$ in the urban plume might be able to explain the smooth increase in low volatility
organic vapors and a higher concentration of organic vapors with a low $\overline{OS_C}$. Box model simulation
showed that the P(OH) were close to the transition regime during three selected periods, indicating
that the elevated $NO_x$ and VOCs in urban plumes can increase the oxidizing capacity. However, the
$P(RO_2)$ was in the VOCs-limited regime, suggesting that the increase in VOCs was mainly
responsible for the daytime enhancement of SOA. Further investigations on the effect of urban
pollutants on SOA formation on the regional scale are still needed for formulating air pollution
control strategies.

*Data availability.* Data from the measurements are available at
https://doi.org/10.6084/m9.figshare.25376059.

*Supplement.* The supplement related to this article is available online at xxx.

*Author contributions.* **MC, YC, and BY** designed the research. **MC, YC, BY, SH, EZ, ZW, YL,**
**TL, WH, WC, QS, WL, YP, BL, QS, and JZ** performed the measurements. **MC, YC, BY, SH,**
**EZ, SY, ZW, YL, TL, WH, WC, QS, WL, YP, BL, QS, and JZ** analyzed the data. **MC, YC, and**
**BY** wrote the paper with contributions from all co-authors.

*Competing interests.* The authors declare that they have no conflict of interest.

*Acknowledgment.* Additional support from the crew of the Heshan supersite and Guangdong
Environmental Monitoring Center is greatly acknowledged.

*Financial support.* This work was supported by the National Key R&D Plan of China (grant no.
2019YFC0214605, 2019YFE0106300, and 2018YFC0213904), the Key-Area Research and
Development Program of Guangdong Province (grant no. 2019B110206001), the National Natural
Science Foundation of China (grant nos. 42305123, 41877302, 91644225, 41775117 and 41807302),
Guangdong Natural Science Funds for Distinguished Young Scholar (grant no. 2018B030306037),
Guangdong Innovative and Entrepreneurial Research Team Program (grant no. 2016ZT06N263),
Guangdong Province Key Laboratory for Climate Change and Natural Disaster Studies (grant no.
2020B1212060025), Guangdong Basic and Applied Basic Research Foundation (grant nos.
2019A151511079, 2019A1515110791, 2023A1515012240, and 2024A1515030221), Science and
Technology Research project of Guangdong Meteorological Bureau (grant no. GRMC2018M07),
the Natural Science Foundation of Guangdong Province, China (grant no. 2016A030311007), the
Research Fund Program of Guangdong-Hongkong-Macau Joint Laboratory of Collaborative
Innovation for Environmental Quality (grant no. GHML2022-005), Science and Technology
Innovation Team Plan of Guangdong Meteorological Bureau (grant no. GRMCTD202003), and
Science and Technology Program of Guangdong Province (Science and Technology Innovation
Platform Category, No. 2019B121201002).

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

**Table 1.** The correlation coefficient between SVOC+LVOC in FIGAERO OA and six OA factors
in AMS OA during different periods.

| | All campaign | Long-range Transport | Urban Air Masses | Coastal Air Masses |
|---|---|---|---|---|
| MOOA | -0.004 | 0.02 | 0.11 | -0.19 |
| LOOA | 0.83 | 0.74 | 0.85 | 0.72 |
| BBSOA | 0.47 | 0.48 | 0.75 | 0.14 |
| HOA | 0.11 | 0.18 | -0.11 | 0.61 |
| BBOA | 0.57 | 0.55 | 0.55 | 0.77 |
| Night-OA | 0.35 | 0.39 | 0.07 | 0.53 |



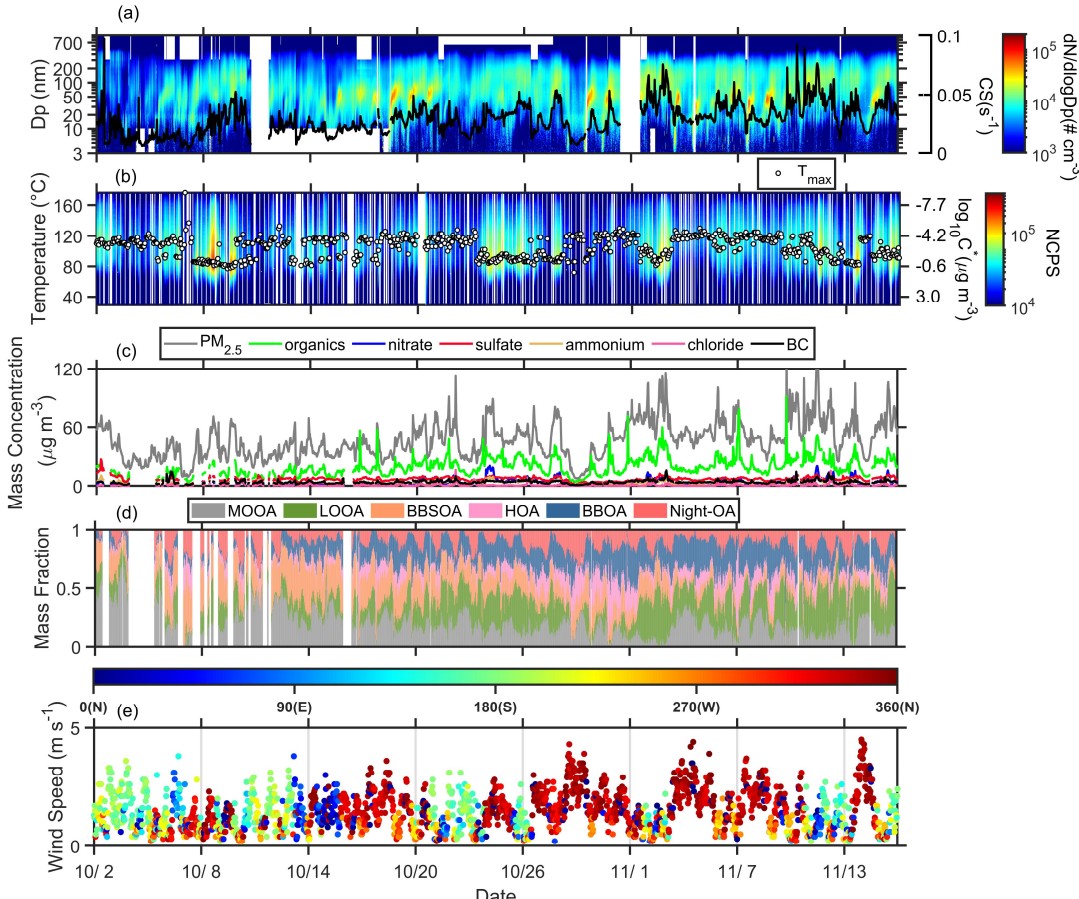


**Figure 1.** Temporal profile of the measured variables during the campaign. (a) particle number size distribution and condensation sink (black line); (b) one-dimensional thermograms of organic compounds (ions containing C, H, and O atoms, referred to as sum thermogram) and the $T_{max}$ values (white dots) measured by the FIGAERO-CIMS; (c) bulk $PM_1$ chemical composition measured by SP-AMS and $PM_1$ concentration; (d) mass fraction of six OA factors from PMF analysis of SP-AMS data; (e) wind speed and wind direction. The color in (b) represents the normalized count per second (ncps) of oxygenated organic compounds calculated based on total count per second (cps) of oxygenated organic compounds at all $m/z$ ($total\ cps$), $m/z$ 127 ($cps_{127}$), and $m/z$ 145 ($cps_{145}$) measured by FIGAERO-I-CIMS, $ncps = \frac{total\ cps}{(cps_{127}+cps_{145})\cdot 10^6}$. The OA factors included more oxygenated OA (MOOA), less oxygenated OA (LOOA), aged biomass burning OA (BBSOA), hydrocarbon-like OA (HOA), biomass burning OA (BBOA), and nighttime OA (night-OA).



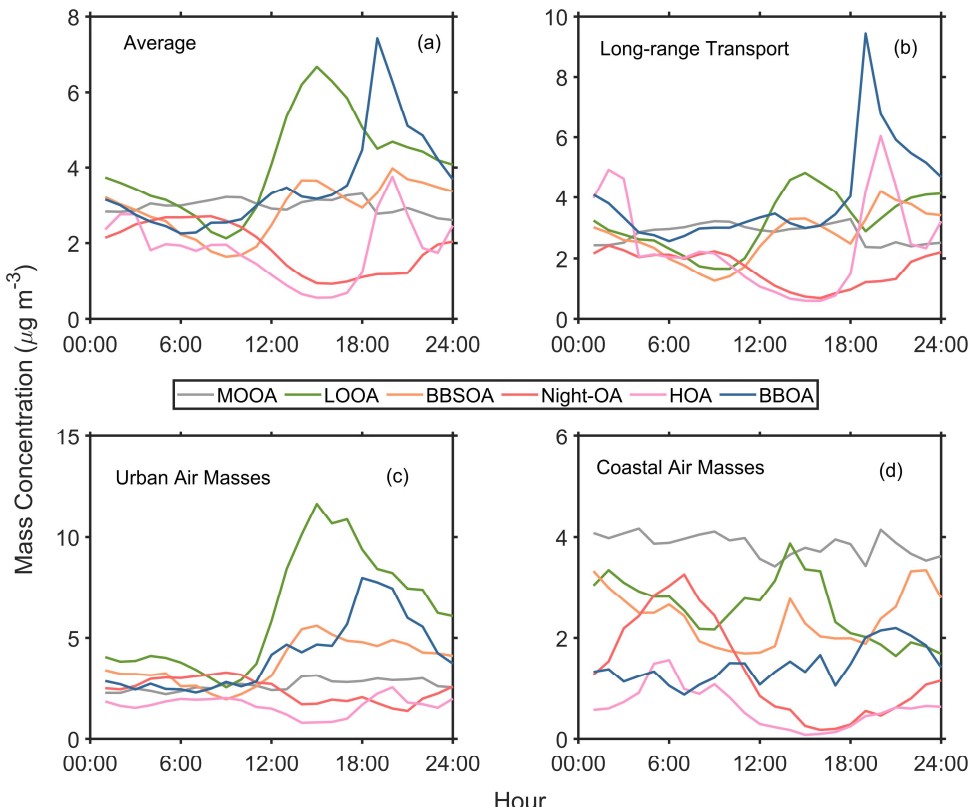


**Figure 2.** Average diurnal variation of six OA PMF factors during (a) the whole campaign, (b)
long-range transport, (c) urban air masses, and (d) coastal air masses periods.

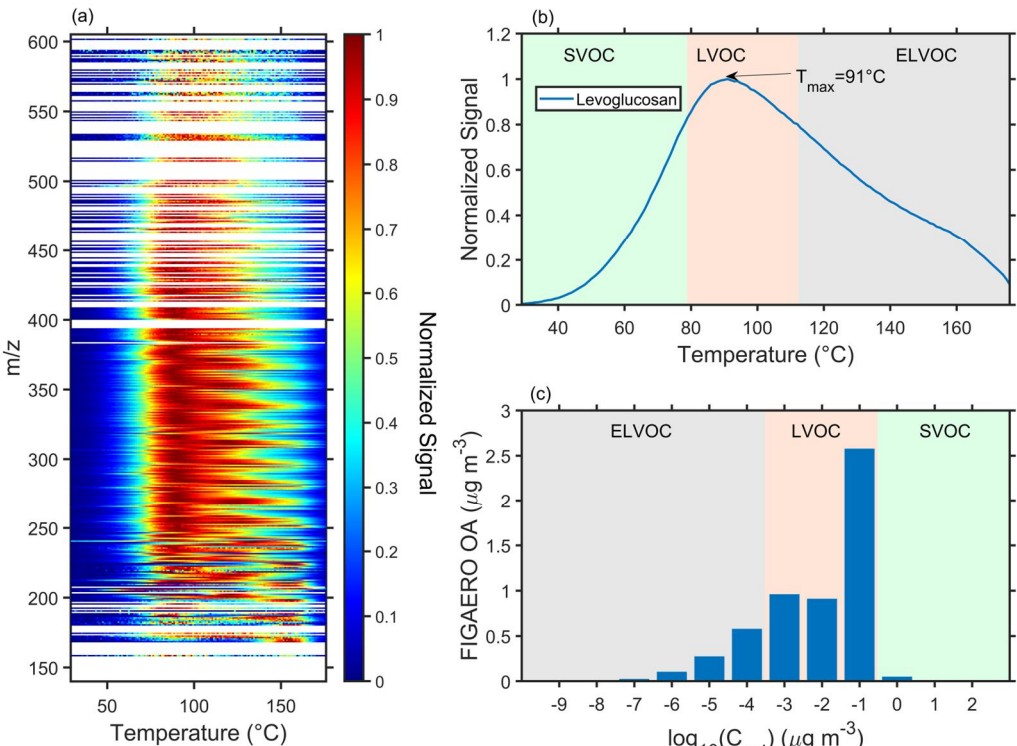


**Figure 3.** The average (a) two-dimensional thermograms of all calibrated and semi-quantified
species, (b) one-dimensional thermogram of levoglucosan, and (c) volatility distribution of all
calibration and semi-quantified species in the particle phase measured by the FIGAERO-CIMS
(referred as FIGAERO OA). The $T_{max}$ was converted to the $C^*$ according to Eqs. (1) and (2).

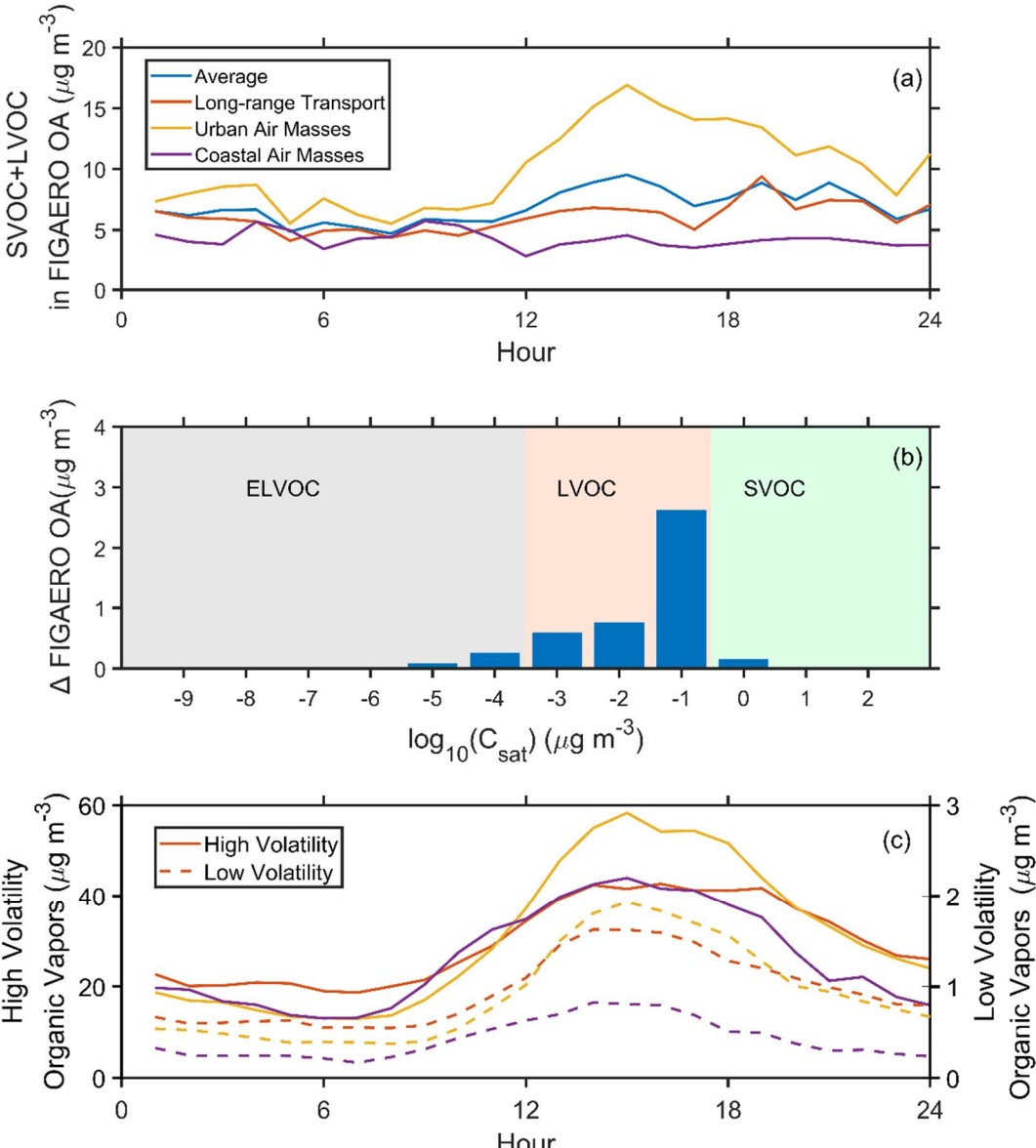


**Figure 4.** Diurnal variation of (a) SVOC+LVOC in FIGAERO OA, (b) the difference of
FIGAERO OA between the urban air masses and long-range transport periods, and (c) low
volatility organic vapors (ELVOC+LVOC, solid lines) and high volatility organic vapors
(SVOC+IVOC+VOC, dash lines) during the whole campaign and three selected periods.

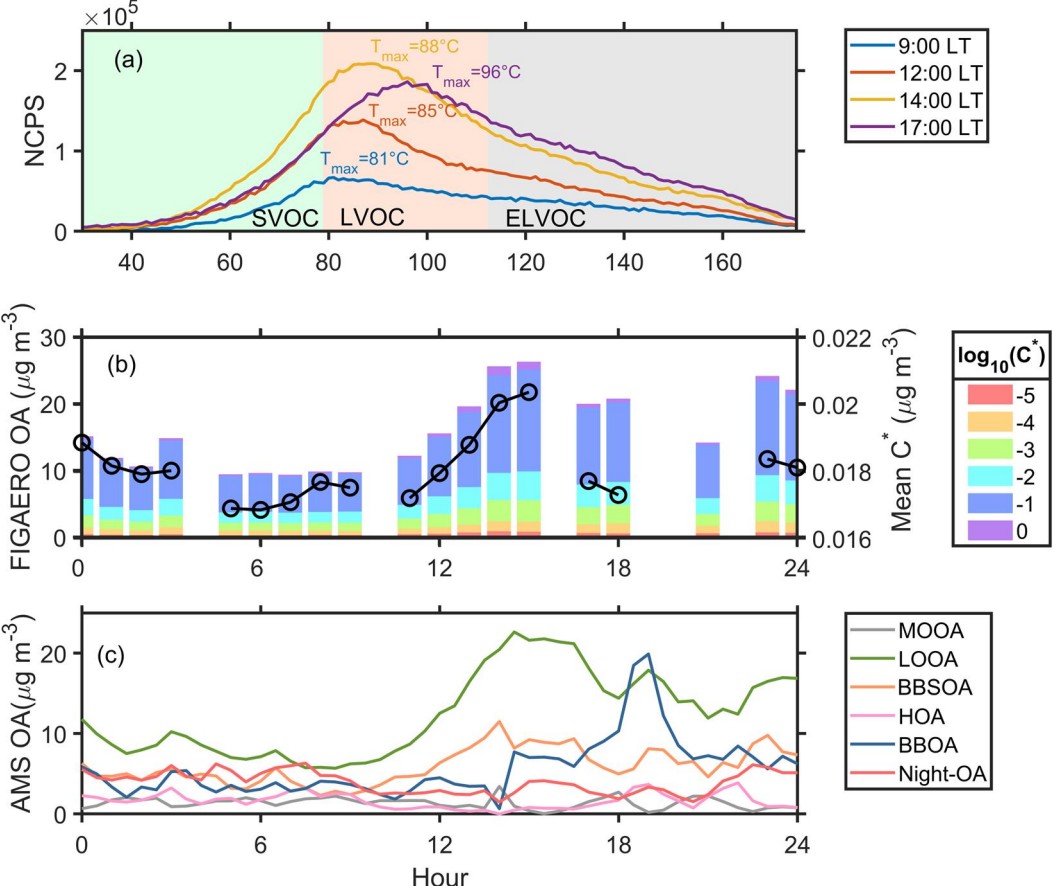


**Figure 5.** (a) The sum thermograms at 9:00, 12:00, 14:00, and 17:00, (b) variation of FIGAERO

OA volatility presented in a volatility range from $10^{-5}$ to $10^0$ µg m$^{-3}$ and mean $C^*$ , and (c)

variation of six OA factors from PMF analysis on 2 November 2019. The mean $C^*(\overline{C^*})$ is

estimated as $\overline{C^*} = 10^{\Sigma f_i log_{10}C_i^*}$, where $f_i$ is the mass fraction of OA with a volatility $C_i^*$.


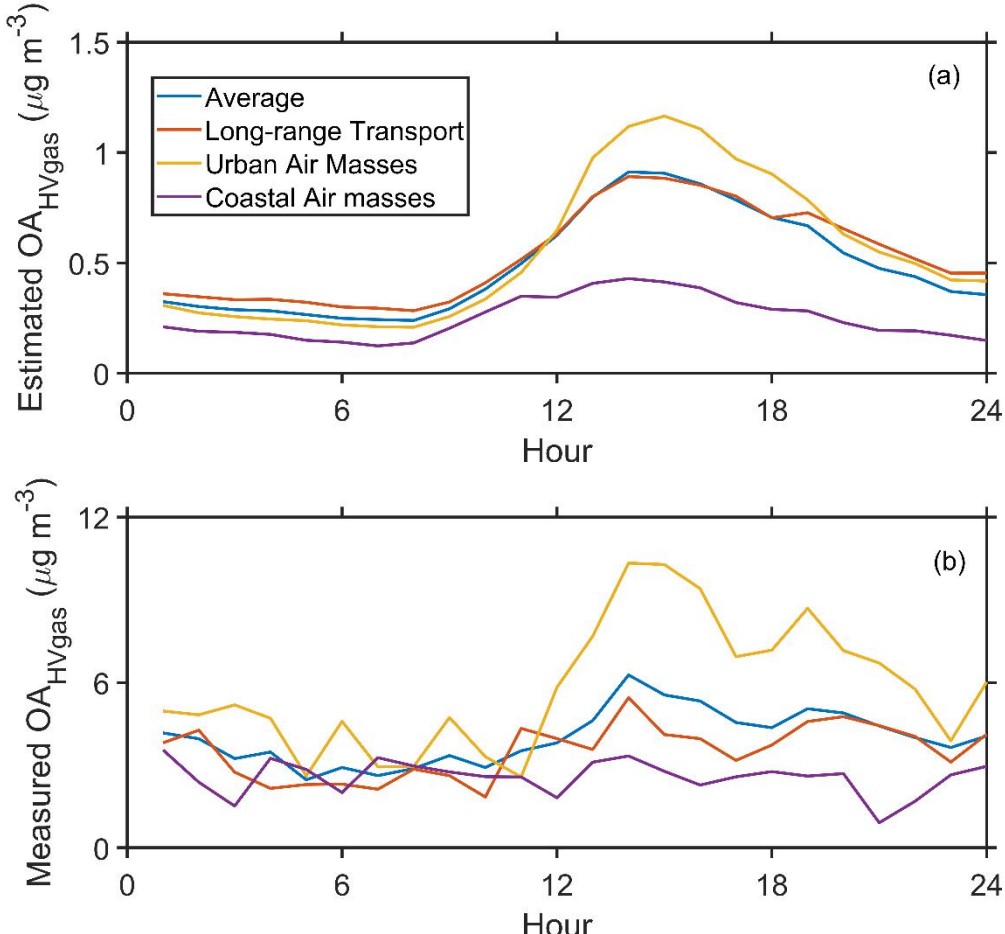


**Figure 6**. The diurnal variation of (a) the estimated contribution of high volatility organic vapors

to the OA (Estimated OA$_{HVgas}$) and (b) the total concentration of corresponding species in the

particles-phase measured by the FIGAERO CIMS.


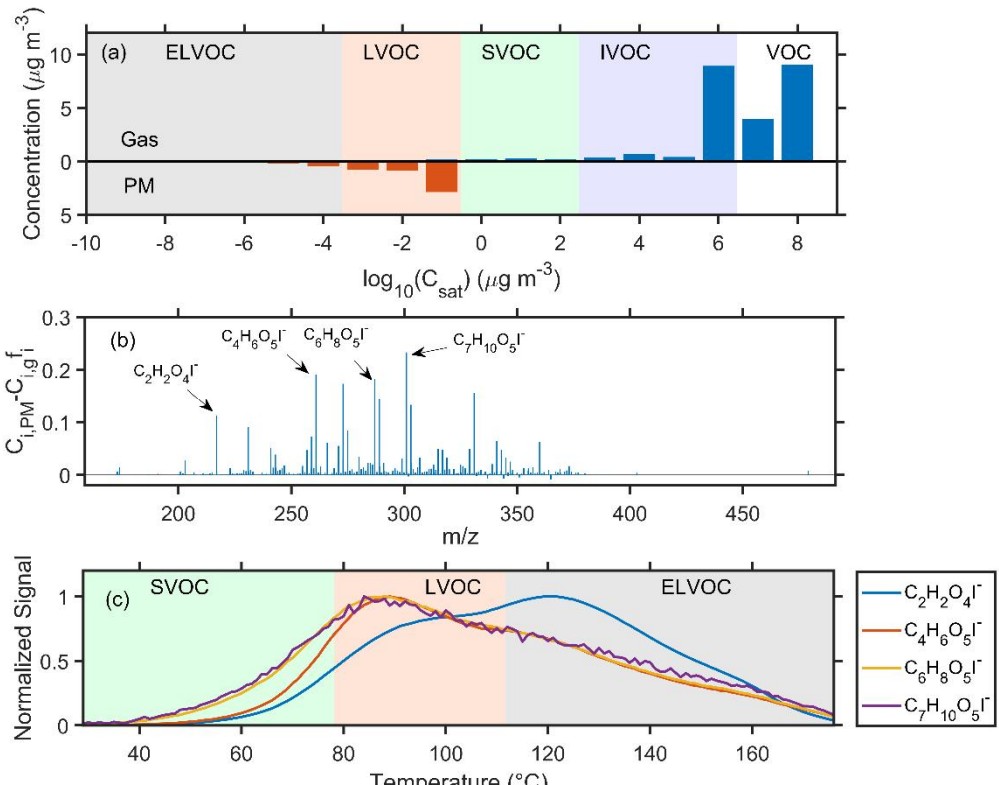


**Figure 7**. (a) The average volatility distribution of high volatility organic vapors in the gas-phase

and particle-phase. (b)The average difference between the measured concentration in the particle-

phase ($C_{i,PM}$) and the estimated concentration ($C_{i,g}f_i$) of different compounds in the high volatility

organic vapors. (c) The average thermograms of $C_2H_2O_4I^-$, $C_4H_6O_5I^-$, $C_6H_8O_5I^-$, and $C_7H_{10}O_5I^-$.

845

846

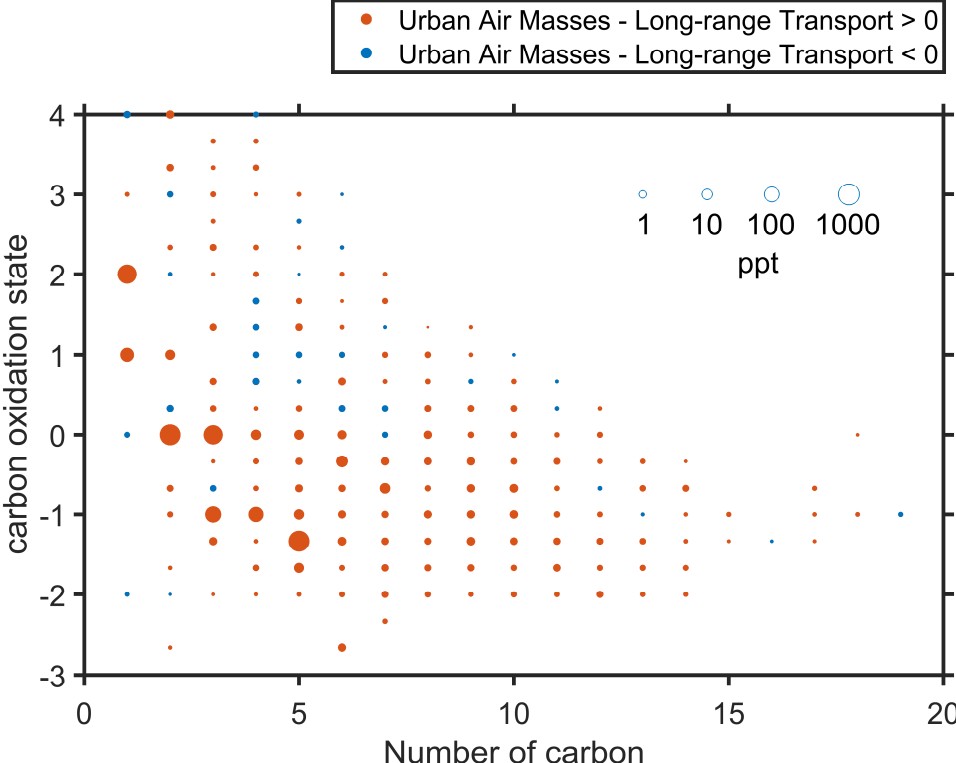

847

**Figure 8.** Difference in the carbon oxidation state ($\overline{OS_C}$) in the gas phase in the afternoon (12:00-

16:00 LT) between the long-range transport and urban air masses periods. The symbol sizes are

proportional to the logarithm of concentration. The symbol colors in a and b represent that the

concentration during the urban air masses period was higher (red) or lower (blue) than that during

the long-range transport period.


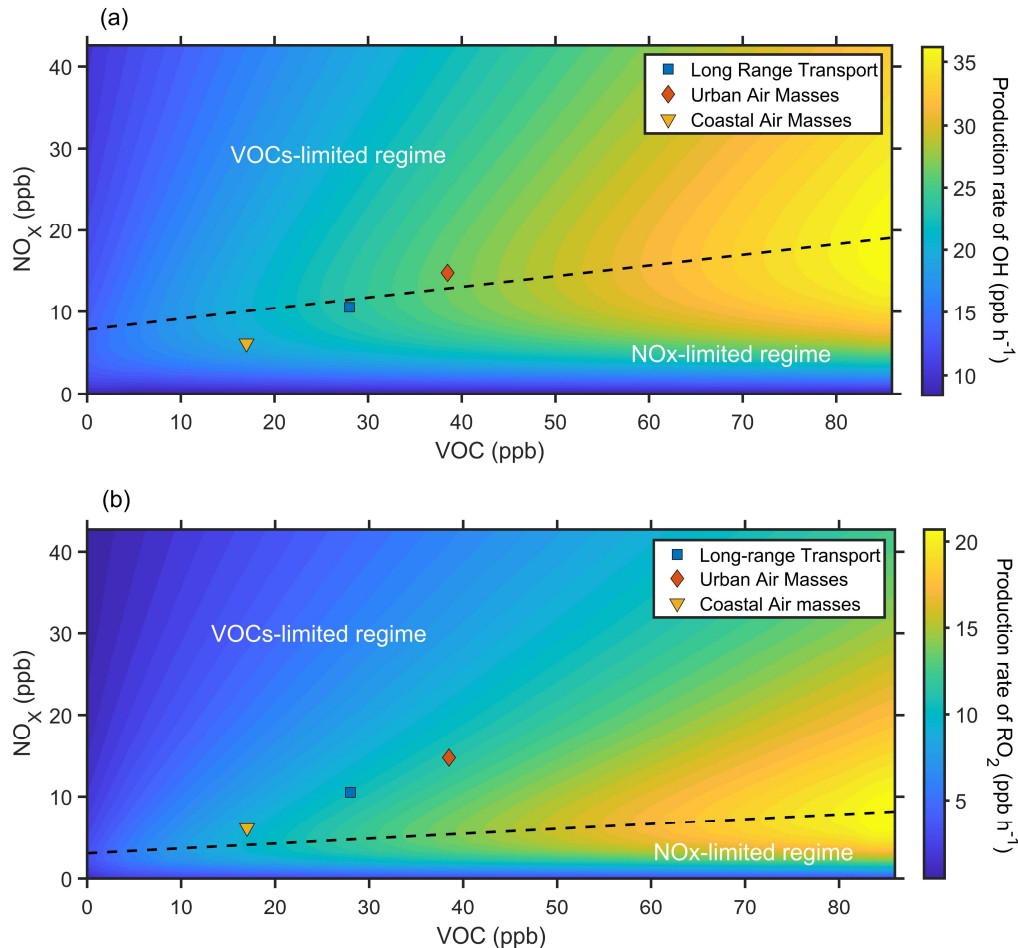


**Figure 9.** The simulated production rate of OH(a) and RO$_2$(b) with NO$_x$ and VOCs concentration
predicted by an observation-constrained box model under campaign average condition. Blue square,
red diamond, and yellow triangle represent the average conditions during long-range transport,
urban air masses, and coastal air masses period, respectively.