# Peer review of "Enhanced daytime secondary aerosol formation driven by"

_EGUsphere, 2024_

## Author Comment (AC1)

This study conducted by Cai et al. demonstrates the significant role of volatile organic compounds (VOCs) from urban plumes in the formation of daytime secondary organic aerosols (SOA) in suburban areas by gas-particle partition through observation using a time-of-flight chemical ionization mass spectrometer coupled with a Filter Inlet for Gases and Aerosol (FIGAERO-CIMS) and other instruments at a suburban site. This manuscript is well-written and fits well to the scope of ACP. I recommend it for publication after the following comments have been addressed.

Major Comments:
1. It is noted that this manuscript utilizes Positive Matrix Factorization (PMF) to distinguish different types of organic compounds. It is necessary to supplement the PMF spectra and diagnosed plot in the supporting information.

Reply: We appreciate the reviewer for this valuable suggestion. A comment from reviewer 3 states that the lower O:C and higher H:C of aBBOA factor is contrary to what it is expected for aging. Our results indicate that this factor was likely formed through oxidation of biomass burning precursors rather the aging process of BBOA. In order to avoid any confusion, we renamed this factor as biomass burning SOA (BBSOA). We added some discussion in section 2.2.2 by providing detailed description of the PMF analysis and PMF spectra in the supporting information.

"The PM$_1$ chemical composition was measured by a soot particle aerosol mass spectrometer (SP-AMS, Aerodyne Research, Inc., USA). The details of the operation and data analysis can be found in Kuang et al. (2021). Source apportionment was performed for organic aerosols in the bulk PM$_1$ using positive matrix factorization (PMF). The organic aerosol could be divided into six components, including two primary OA factors and four secondary OA factors. The mass spectral profiles of six OA factors are shown in Figure S3. The timeseries and diurnal variation of these factors are presented in Figure S4.

The primary OA factors include hydrocarbon-like OA (HOA), mainly contributed by traffic and cooking emissions and biomass burning OA (BBOA) originating from biomass burning combustion. The HOA was identified by hydrocarbon ions $C_xH_y^+$. Owing to the prominent hydrocarbon ions and low O:C value (0.10), HOA could be attributed to primary emission from cooking and traffic. The BBOA was recognized by the markers $C_2H_4O_2^+$ (m/z 60.022, 0.5%) and $C_3H_5O_2^+$ (m/z 73.029, 0.4%), which are considered tracers for biomass burning OA (Ng et al., 2011).

The SOA factors include biomass burning SOA (BBSOA) likely formed from oxidation of biomass burning emission, less oxygenated OA (LOOA) provided by strong daytime photochemical formation, more oxygenated OA (MOOA) related to regional transport, and nighttime-formed OA (Night-OA) contributed by secondary formation during nighttime. The BBSOA was likely formed through oxidation of biomass burning precursors, which was supported by the evening peak at about 19:00 LT (Fig. S4). BBSOA showed a similar variation trend with $C_6H_2NO_4^+$, which might be contributed by oxidation of gaseous precursors from biomass burning emissions (Wang et al., 2019; Bertrand et al., 2018). The significant afternoon peak of LOOA indicates its formation through photochemical reactions, which would be detailly discussed in section 3.1. The negligible diurnal

variation and the highest O:C value (1.0) of MOOA suggested that it could be aged OA resulting from long-range transport. Night-OA was formed through NO₃ nighttime chemistry, supported by a pronounced evening elevation and positive correlation with nitrate (R=0.67).The detailed determination of PMF factors has been found in Kuang et al. (2021) and Luo et al. (2022).

[Figure]

Figure S3. Mass spectral profile of six OA factors. The colors represent different family groups.

[Figure]

Figure S4. Timeseries and diurnal variation of six OA factors.
"

2. Figure 6 illustrates the difference in carbon oxidation state () between different periods. However, the description of Figure 6 in the text is insufficient. The authors should provide further explanation on Figure 6, detailing the observed differences in organic aerosol states in different carbon numbers during various periods.

Reply: We appreciate the reviewer for this suggestion. We have added some discussion in Line 433-436:

"This trend was more significant for carbon number between 2 and 5, indicating a higher concentration of small molecules with low $\overline{OS_C}$ during urban air masses period. The $\overline{OS_C}$ of major $C_5$ compounds was about -1.33, which was mainly contributed by $C_5H_8NOI^-$, highlighting the role of $NO_x$ chemistry."

3.  Figure 7 present the different production of OH and $RO_2$ in different VOC and $NO_X$ Adding the boundary of VOCs and NOx limited region could help reader better understanding the change of production of OH and $RO_2$ in different periods.

Reply: We appreciate the reviewer for this suggestion. We have modified figure 7 (now figure 9) by adding the boundary of VOCs and $NO_x$ control regime.

[Figure]

Figure 9. The simulated production rate of OH(a) and $RO_2$(b) with NOx and VOCs concentration predicted by an observation-constrained box model under campaign average condition. Blue square, red diamond, and yellow triangle represent the average conditions during long-range transport, urban air masses, and coastal air masses period, respectively.

4.  Line 243: The author mentions that the correlation between LOOA concentration and particle surface area suggests a relationship between gas-particle partitioning and LOOA formation. However, particle mass concentration is also positively correlated with the particle surface area. The authors should provide more evidence about the contribution of gas-particle partitioning on LOOA formation.

Reply: We thank the reviewer for this suggestion. We agree that particle mass concentration is positively correlated with the particle surface area. The positive relationship between the LOOA and particle surface area might not be strong evidence of the contribution of gas-particle partitioning. We further analysis the relationship between organic vapors and six OA factors. It was shown that

the LOOA was highly correlated with organic vapors (R=0.84), while we did not observe such high correlation for other OA factors. These results suggest that the formation of LOOA could be contributed by gas-particle partitioning. We have modified the corresponding sentences in Line 292-296:

"LOOA showed a noticeable increase corresponding to the particle surface area (Fig. S8), while we did not observe such correlation for other SOA factors (MOOA and BBSOA). Furthermore, LOOA exhibited a stronger positive correlation with organic vapors measured by the FIGAERO-CIMS compared to other OA factors (Fig. S9). These results suggested that the daytime formation of LOOA was attributed to gas-particle partitioning."

5. Line 367: In urban plumes, the production rate of OH increases with the increase of NOx and VOC, transitioning to the VOC-limited regime. Why is it stated that NOx suppressed the production rate of OH at this period?

Reply: We are grateful to the reviewer for this valuable suggestion. Compared with the period influenced by coastal air masses, the P(OH) in urban plumes was elevated owing to the increase of VOCs and $NO_x$. However, the sensitivity regime of P(OH) shifted towards being VOCs-limited during the urban air masses period. It suggests that the P(OH) might be suppressed with further increases in $NO_x$. To avoid any confusion, this sentence (Line 456-458) has been revised to:

"Interestingly, the sensitivity regime of P(OH) changed to the VOCs-limited during the urban air masses period, suggesting that the production of OH would be suppressed with continued increases in $NO_x$."

Specific comments:

1. Line 259: It should be "long-range transport".
   Reply: It has been revised.

2. Figure S8: please provide R value.
   Reply: The R value has been added.

[Figure]

Figure S10. Relationship between odd-oxygen ($O_X$, $O_X=O_3+NO_2$) and the concentration of organic vapors measured by the FIGAERO-CIMS in the afternoon (10:00-16:00 LT).

3. Line 389: It should be "dramatic".
Reply: It has been revised.

**Reference**

Bertrand, A., Stefenelli, G., Jen, C. N., Pieber, S. M., Bruns, E. A., Ni, H., Temime-Roussel, B., Slowik, J. G., Goldstein, A. H., El Haddad, I., Baltensperger, U., Prévôt, A. S. H., Wortham, H., and Marchand, N.: Evolution of the chemical fingerprint of biomass burning organic aerosol during aging, Atmos. Chem. Phys., 18, 7607-7624, 10.5194/acp-18-7607-2018, 2018.
Kuang, Y., Huang, S., Xue, B., Luo, B., Song, Q., Chen, W., Hu, W., Li, W., Zhao, P., Cai, M., Peng, Y., Qi, J., Li, T., Wang, S., Chen, D., Yue, D., Yuan, B., and Shao, M.: Contrasting effects of secondary organic aerosol formations on organic aerosol hygroscopicity, Atmos. Chem. Phys., 21, 10375-10391, 10.5194/acp-21-10375-2021, 2021.
Luo, B., Kuang, Y., Huang, S., Song, Q., Hu, W., Li, W., Peng, Y., Chen, D., Yue, D., Yuan, B., and Shao, M.: Parameterizations of size distribution and refractive index of biomass burning organic aerosol with black carbon content, Atmos. Chem. Phys., 22, 12401-12415, 10.5194/acp-22-12401-2022, 2022.
Ng, N. L., Canagaratna, M. R., Jimenez, J. L., Zhang, Q., Ulbrich, I. M., and Worsnop, D. R.: Real-Time Methods for Estimating Organic Component Mass Concentrations from Aerosol Mass

Spectrometer Data, Environmental Science & Technology, 45, 910-916, 10.1021/es102951k, 2011.

Wang, Y., Hu, M., Wang, Y., Zheng, J., Shang, D., Yang, Y., Liu, Y., Li, X., Tang, R., Zhu, W., Du, Z., Wu, Y., Guo, S., Wu, Z., Lou, S., Hallquist, M., and Yu, J. Z.: The formation of nitro-aromatic compounds under high NOx and anthropogenic VOC conditions in urban Beijing, China, Atmos. Chem. Phys., 19, 7649-7665, 10.5194/acp-19-7649-2019, 2019.

---

## Author Comment (AC2)

Cai et al discuss the enhancement of secondary organic aerosol downwind of urban centers due to increased partitioning of semi-volatile vapors. FIGAERO-CIMS is employed to assess the volatility evolution of particulate species over time. PMF is applied to SP-AMS data in order to understand the sources of OA. Overall the work is thorough and the limitations are clearly stated. I think the content of the work is appropriate for ACP and therefore I would recommend publication after the following comments are addressed.

1. I greatly appreciate the transparency of the PMF factors being provided in the SI, however, more description of these factors and how they were determined is required in the main text. Each factor should be described individually and it should be discussed why it was attributed to the specific source it was. Particularly, the Night-OA, BBOA, and aBBOA factors seem visually quite similar and a discussion of the specific difference would be very helpful. Additionally the average composition values for HOA and aBBOA are identical which seems surprising.

Reply: We appreciate the reviewer for this valuable suggestion. The composition values for HOA are copy mistake and has been revised. A comment from reviewer 3 states that the lower O:C and higher H:C of aBBOA factor is contrary to what it is expected for aging. Our results indicate that this factor was likely formed through oxidation of biomass burning precursors rather the aging process of BBOA. To avoid any confusion, we renamed this factor as biomass burning SOA (BBSOA). Additionally, we have modified section 2.2.2 by providing more description of these factors.

"The PM$_1$ chemical composition was measured by a soot particle aerosol mass spectrometer (SP-AMS, Aerodyne Research, Inc., USA). The details of the operation and data analysis can be found in Kuang et al. (2021). Source apportionment was performed for organic aerosols in the bulk PM$_1$ using positive matrix factorization (PMF). The organic aerosol could be divided into six components, including two primary OA factors and four secondary OA factors. The mass spectral profiles of six OA factors are shown in Figure S3. The timeseries and diurnal variation of these factors are presented in Figure S4.

The primary OA factors include hydrocarbon-like OA (HOA), mainly contributed by traffic and cooking emissions and biomass burning OA (BBOA) originating from biomass burning combustion. The HOA was identified by hydrocarbon ions $C_xH_y^+$. Owing to the prominent hydrocarbon ions and low O:C value (0.10), HOA could be attributed to primary emission from cooking and traffic. The BBOA was recognized by the markers $C_2H_4O_2^+$ (m/z 60.022, 0.5%) and $C_3H_5O_2^+$ (m/z 73.029, 0.4%), which are considered tracers for biomass burning OA (Ng et al., 2011).

The SOA factors include biomass burning SOA (BBSOA) likely formed from oxidation of biomass burning emission, less oxygenated OA (LOOA) provided by strong daytime photochemical formation, more oxygenated OA (MOOA) related to regional transport, and nighttime-formed OA (Night-OA) contributed by secondary formation during nighttime. The BBSOA was likely formed through oxidation of biomass burning precursors, which was supported by the evening peak at about 19:00 LT (Fig. S4). BBSOA showed a similar variation trend with $C_6H_2NO_4^+$, which might be contributed by oxidation of gaseous precursors from biomass burning emissions (Wang et al., 2019; Bertrand et al., 2018). The significant afternoon peak of LOOA indicates its formation through photochemical reactions, which would be detailly discussed in

section 3.1. The negligible diurnal variation and the highest O:C value (1.0) of MOOA suggested that it could be aged OA resulting from long-range transport. Night-OA was formed through $NO_3$ nighttime chemistry, supported by a pronounced evening elevation and positive correlation with nitrate (R=0.67).The detailed determination of PMF factors has been found in Kuang et al. (2021) and Luo et al. (2022).

[Figure]

Figure S3. Mass spectral profile of six OA factors. The colors represent different family groups.

[Figure]

Figure S4. Timeseries and diurnal variation of six OA factors.
"

2. This is likely beyond the scope of this study, but I wonder if any consideration was given to using PMF to identify different sources or formation pathways from the FIGAERO-CIMS data as in Buchholz et al as well as other studies. Perhaps just an idea for future work.

Reply: We are grateful to the reviewer for this valuable suggestion. Buchholz et al. (2020) employed PMF to the FIGAERO-CIMS data in the laboratory study. However, performing PMF analysis to the campaign thermograms data is challenging, since the amount of data is huge. After a lot of effort, we have successfully conducted a PMF analysis of the campaign thermograms data. The 20s averaged data were input into the PMF model. The data processing method, such as error schemes, was based on Buchholz et al. (2020). An example of our results can be found in the following figure (fig. 2.1). We are still analyzing our results; we hope to publish them soon.

[Figure]

Figure 2.1. The (a) PMF thermogram factors and (b) desorption temperature against data point index.

3. Line 168: A MW of 200 g mol$^{-1}$ is assumed, but can you not retrieve an average MW from the CIMS data? Would it be biased too high due to low detection efficiency of less oxidized, low MW species?

Reply: We appreciate the reviewer for this valuable suggestion. We calculated the average molecular weight based on the CIMS data. It shows that the average molecular weight of particle-phase compounds was 194.9 g mol$^{-1}$, which was close to the assumed $M_w$ (200 g mol$^{-1}$). We noticed that using a universal $M_w$ value in the $C^*$ estimation of each compound could lead to a deviation. Ylisirniö et al. (2021) calculated the $C^*$ based on the $M_w$ of the compound determined

by the FIGAERO-CIMS, we believed it would be a better way in calculating the $C^*$. The comparison of the estimated $C^*$ based on the universal $M_w$ and actual $M_w$ was shown in the following figure (Fig. 3.1). It suggested that adopting a universal $M_w$ value (200 g mol⁻¹) could lead to an overestimate of $C^*$ for compounds with a $M_w$ lower than 200 g mol⁻¹, while this trend was overturned for species was a $M_w$ higher than 200 g mol⁻¹.

[Figure]

Figure 3.1. The estimated $C^*$ based on the universal $M_w$ and actual $M_w$.

Thus, we used the $M_w$ of the compound in eq. (2) and modified the corresponding discussion and figures in our manuscript.

Line 348-352, "Table 1 investigated the relationship between SVOC+LVOC and six OA factors. The SVOC+LVOC in FIGAERO OA had a significant positive correlation (R=0.72-0.85) with the LOOA, especially during the urban air masses period (R=0.85, Fig. S14 and Table 1), suggesting that the LOOA formation was mainly responsible for the increase of OA volatility.

**Table 1.** The correlation coefficient between SVOC+LVOC in FIGAERO OA and six OA factors in AMS OA during different periods.

|          | All campaign | Long-range Transport | Urban Air Masses | Coastal Air Masses |
|----------|--------------|----------------------|------------------|--------------------|
| MOOA     | -0.004       | 0.02                 | 0.11             | -0.19              |
| LOOA     | 0.83         | 0.74                 | 0.85             | 0.72               |
| BBSOA    | 0.47         | 0.48                 | 0.75             | 0.14               |
| HOA      | 0.11         | 0.18                 | -0.11            | 0.61               |
| BBOA     | 0.57         | 0.55                 | 0.55             | 0.77               |
| Night-OA | 0.35         | 0.39                 | 0.07             | 0.53               |

[Figure]

Figure S14. Relationship between the SVOC+LVOC in FIGAERO OA and LOOA in AMS OA during (a) the whole campaign, (b) long-range transport, (c) urban air masses, and (d) coastal air masses periods.

"

Figure 5,

"

[Figure]

**Figure 5.** (a) The sum thermograms at 9:00, 12:00, 14:00, and 17:00, (b) variation of FIGAERO OA volatility presented in a volatility range from $10^{-5}$ to $10^{0}$ μg m$^{-3}$ and mean $C^{*}$ , and (c) variation of six OA factors from PMF analysis on 2 November 2019. The mean $C^{*}(\overline{C^{*}})$ is estimated as $\overline{C^{*}} = 10^{\Sigma f_i log_{10} C_i^{*}}$, where $f_i$ is the mass fraction of OA with a volatility $C_i^{*}$.

"

4. The determination of the volatility of gas-phase species is based on the formation pathway of the species. This is important to consider due to the different functional groups likely to be dominant in products formed via autoxidation and I applaud the authors consideration of this. Given the uncertainties associated with the volatility estimation, the method employed in this study is likely good enough, however, as explicit determination of the pathway of formation is impossible, some discussion of this limitation should be added. I also think a sentence describing how H:C and O:C (Fig S6) were used to determine the pathway of formation as well as relevant references in the main text would be helpful.

Reply: We appreciate the reviewer for this valuable suggestion. We have modified the corresponding sentences by adding some discussion about the limitations and how to determine the pathway in the main text. Also, we have revised the slope of black line in the Van-Krevelen diagram, since the previous one was a copy mistake.

Line 214-218, "For gas-phase organic compounds (organic vapors), we first divided them into two groups based on their potential oxidation pathways (multi-generation OH oxidation and autoxidation, solid line in Fig. S7) and then used different parameters in their volatility estimation. The classification of pathways was based on the molecular characteristics of oxidation products of aromatics and monoterpene, respectively (Wang et al., 2020).

[Figure]

Figure S7. Van-Krevelen diagram (O/C ratio versus H/C ratio) of gas-phase organic compounds measured by FIGAERO-CIMS. The symbol size is proportional to the mass concentration of organic vapors and the color code represents the volatility. The black solid line divided the organic vapors potentially formed through the autoxidation pathway (upper regime) and multi-generation OH oxidation pathway (lower regime), based on the oxidation products aromatics and monoterpene, respectively (Wang et al., 2022; Wang et al., 2020).

"

Line 225-227, "It should be noted that this method can only roughly distinguish the formation pathways of ambient organic vapors, since it is based on the oxidation products of specific species in a laboratory study"."

5.  I am confused about the assignment of species with $C^* > 10^{0.5}$ as "non-condensable." This boundary is within the SVOC VBS range and particularly under the high mass loadings one

could see even downwind of urban plumes, it seems species with higher volatilities may contribute substantially to the particle phase. Is this determination specific to the conditions of this study in some way or based on other literature?

Reply: We appreciate the reviewer for this valuable suggestion. Nie et al. (2022) calculated the contribution of condensing organic vapor to the formation of SOA. For organic vapor with relatively lower volatility ($C^*$ ≤0.01 μg m$^{-3}$), the condensation to particle-phase was regarded as irreversible. Wang et al. (2022) integrated organic vapor from the lowest volatility bin to $C^*$ ≤ $10^{0.5}$ μg m$^{-3}$ and regard them as condensable vapors. Our assignment of organic vapors was based on Wang et al. (2022). We noticed that using the name "non-condensable organic vapors" and "condensable organic vapors" could lead to confusion, since "non-condensable organic vapors" can also reach the particle phase through gas-particle partitioning. Thus, we modified the classification, ELVOC and LVOC are classified as low volatility organic vapors ($C^*$ ≤ 0.3 μg m$^{-3}$), while SVOC, IVOC and VOC fall into another category regarded as high volatility organic vapors ($C^*$ > 0.3 μg m$^{-3}$). The corresponding sentences and figures in the manuscript have been revised.

6. While stated correctly in the text, I think the boundary of the SVOC class is incorrect in Fig 4b. SVOCs should extend to $10^{-0.5}$ not $10^{0.5}$ ug m$^{-3}$, assuming these are C*(300 K).

Reply: We are grateful to the reviewer for this valuable suggestion. We have revised the boundary of the SVOC in fig. 3, fig. 4, and fig. S12.

"

[revised manuscript text omitted]

---

## Author Comment (AC3)

The work by Cai et al. investigates the SOA formation in downwind regions of urban areas, focusing on the PRD region of China in the fall of 2019. The FIGAERO-CIMS was employed to analyze the molecular composition and volatility of organic compounds in both gas and particle phases. The findings highlight significant daytime SOA formation driven by gas-particle partitioning, influenced by urban pollutants such as NOx and volatile organic compounds (VOCs). The paper is well-structured, clearly written, and a valuable contribution to the field of atmospheric sciences, particularly in understanding the dynamics of SOA formation in urban-influenced suburban areas. With the following comments addressed, it would be suitable for publication in ACP.

1. The aBBOA factor appears to have a lower O:C and a higher H:C compared to the BBOA factor (Figure S3). This is contrary to what it is expected for aging. This makes me wonder how these PMF factors were exactly assigned. Some explanation will be helpful.

Reply: We appreciate the reviewer for this valuable suggestion. The lower O:C ratio and higher H:C ratio implies that aBBOA was likely formed through oxidation of biomass burning precursors rather the aging process of BBOA. To avoid any confusion, we rename this factor as biomass burning SOA (BBSOA). We have modified section 2.2.2 by providing more description of these factors.

"The PM$_1$ chemical composition was measured by a soot particle aerosol mass spectrometer (SP-AMS, Aerodyne Research, Inc., USA). The details of the operation and data analysis can be found in Kuang et al. (2021). Source apportionment was performed for organic aerosols in the bulk PM$_1$ using positive matrix factorization (PMF). The organic aerosol could be divided into six components, including two primary OA factors and four secondary OA factors. The mass spectral profiles of six OA factors are shown in Figure S3. The timeseries and diurnal variation of these factors are presented in Figure S4.

The primary OA factors include hydrocarbon-like OA (HOA), mainly contributed by traffic and cooking emissions and biomass burning OA (BBOA) originating from biomass burning combustion. The HOA was identified by hydrocarbon ions $C_xH_y^+$. Owing to the prominent hydrocarbon ions and low O:C value (0.10), HOA could be attributed to primary emission from cooking and traffic. The BBOA was recognized by the markers $C_2H_4O_2^+$ (m/z 60.022, 0.5%) and $C_3H_5O_2^+$ (m/z 73.029, 0.4%), which are considered tracers for biomass burning OA (Ng et al., 2011).

The SOA factors include biomass burning SOA (BBSOA) likely formed from oxidation of biomass burning emission, less oxygenated OA (LOOA) provided by strong daytime photochemical formation, more oxygenated OA (MOOA) related to regional transport, and nighttime-formed OA (Night-OA) contributed by secondary formation during nighttime. The BBSOA was likely formed through oxidation of biomass burning precursors, which was supported by the evening peak at about 19:00 LT (Fig. S4). BBSOA showed a similar variation trend with $C_6H_2NO_4^+$, which might be contributed by oxidation of gaseous precursors from biomass burning emissions (Wang et al., 2019; Bertrand et al., 2018). The significant afternoon peak of LOOA indicates its formation through photochemical reactions, which would be detailly discussed in section 3.1. The negligible diurnal variation and the highest O:C value (1.0) of MOOA suggested that it could be aged OA resulting from long-range transport. Night-OA was formed through NO$_3$ nighttime chemistry, supported by

a pronounced evening elevation and positive correlation with nitrate (R=0.67).The detailed determination of PMF factors has been found in Kuang et al. (2021) and Luo et al. (2022).

[Figure]

Figure S3. Mass spectral profile of six OA factors. The colors represent different family groups.

[Figure]

Figure S4. Timeseries and diurnal variation of six OA factors.

"

2. Line 179-183: There does not seem to be a clear trend between mass loading and Tmax and the calibration mass loading range does not cover the campaign mass loading center (Figure S5). Can the authors explain the rationale of picking the fitting parameters of the experiment with Dp 200 nm and mass loading = 407 ng rather than for example the parameters from fitting all experiments? What is the direction of bias introduced by this choice?

Reply: We appreciate the reviewer for this valuable suggestion. Wang and Hildebrandt Ruiz (2018) indicated that the relationship between mass loading and $T_{max}$ can be described by a sigmoid function. The non-monotonic trend between mass loading and $T_{max}$ could partly owing to the fact that the mass loading reached the "plateau" region in the sigmoid function. We also performed the

$T_{max}$ calibration based on the syringe deposition method. Our results suggest that the $T_{max}$ value does not always increase with the increase in mass loading (fig. 2.1). Huang et al. (2018) suggested that the non-linear correlation between $T_{max}$ shift and mass loading might be due to matrix or saturation effects. Considering the $T_{max}$ dependence might reach a plateau, the increase in mass loading might play a minor effect in our calibration results. Thus, we did not perform any further experiments with higher mass loading. The mass loading and average particle volume size distribution (PVSD) shows that the mass loading centered at about 602 ng and the PVSD centered at about 400 nm. However, generating particles larger than 250 nm is challenging for our atomizer. Thus, the experiment with a Dp of 200 nm and mass loading of 407 ng were utilized because mass loading and diameter are the closest to the ambient samples.

We added some discussion about this phenomenon and choosing the fitting parameters in line 205-213,

"Note that the $T_{max}$ can vary with mass loading, and it is necessary to consider for estimation the relationship between $T_{max}$ and $C^*$ (Wang and Hildebrandt Ruiz, 2018). Our calibration results demonstrated that the correlation between $T_{max}$ shift and mass loading was not linear, which may be attributed to matrix or saturation effects (Huang et al., 2018). During the measurement, the collected mass loading centered at about 620 ng and the particle volume size distribution (PVSD) centered at about 400 nm (Fig. S6). Thus, the fitting parameters (a=-0.206 and a=3.732) of the calibration experiment with a diameter of 200 nm and mass loading of 407 ng were adopted in the $C^*$ calculation, since the mass loading and diameter are the closest to the ambient samples."

[Figure]

Figure 2.1 Thermograms for different compounds at different loading conditions.

3. Line 184-186: it would be helpful to describe how the black line in Figure S6 that differentiates the oxidation pathways was determined in light of existing literature in a sentence or two.

Reply: We thank the reviewer for this valuable suggestion. We have added some descriptions about how to determine the pathway in the main text. Also, we have revised the slope of black line in the Van-Krevelen diagram, since the previous one was a copy mistake.

Line 214-218, "For gas-phase organic compounds (organic vapors), we first divided them into two groups based on their potential oxidation pathways (multi-generation OH oxidation and autoxidation, solid line in Fig. S7) and then used different parameters in their volatility estimation. The classification of pathways was based on the molecular characteristics of oxidation products of aromatics and monoterpene, respectively (Wang et al., 2020).

[Figure]

Figure S7. Van-Krevelen diagram (O/C ratio versus H/C ratio) of gas-phase organic compounds measured by FIGAERO-CIMS. The symbol size is proportional to the mass concentration of organic vapors and the color code represents the volatility. The black solid line divided the organic vapors potentially formed through the autoxidation pathway (upper regime) and multi-generation OH oxidation pathway (lower regime), based on the oxidation products aromatics and monoterpene, respectively (Wang et al., 2022; Wang et al., 2020). "

Line 225-227, "It should be noted that this method can only roughly distinguish the formation pathways of ambient organic vapors, since it is based on the oxidation products of specific species in a laboratory study."

4. Line 217-220: These observation data used to constrain F0AM simulations were not mentioned in the instrumentation section of the paper. Are these collocated and published data? Adding a brief description would provide necessary context.

Reply: We thank the reviewer for this valuable comment. These observation data were measured by a series of instruments during the campaign. The background concentration of CH₄ was set as 1.8 ppm (Wang et al., 2011). We added a brief description of the corresponding instruments in the instrumentation section.

Line 172-180,

 "**2.2.4 Other parameters**

The non-methane hydrocarbons (NMHC) were measured by an online GC-MS-FID (Wuhan Tianhong Co., Ltd, China). The concentration of oxygenated VOCs, including formaldehyde (HCHO) and acetaldehyde (CH₃CHO), were measured using high-resolution proton transfer reaction time-of-flight mass spectrometry (PTR-ToF-MS, Ionicon Analytik, Austria). HONO was detected by the gas and aerosol collector (GAC) instrument (Dong et al., 2012). Trace gases, including $O_3$, $NO_x$, and CO, were measured by gas analyzers (model 49i, 42i, and 48i, Thermo Scientific, US). Meteorological parameters (i.e., wind speed, wind direction, and temperature) were measured by a weather station (Vantage Pro 2, Davis Instruments Co., US). "

We also modified the sentence in Line 265-268,

"The simulation was constrained with the observation data of non-methane hydrocarbons (NMHC), HCHO, CH₃CHO, NO, CO, HONO, and meteorological parameters (RH, temperature, photolysis rates, and pressure). The background concentration of CH₄ was set as 1.8 ppm (Wang et al., 2011)."

5. Line 302: The term "non-condensable" ($C* > 10^{0.5}$ µg m⁻³) is a bit confusing. These vapors are apparently condensable SVOCs that would partition between gas and particle phases. Is this definition based on specific literature? Clarifying this term would enhance understanding.

Reply: We appreciate the reviewer for this valuable suggestion. Our assignment of organic vapors was based on Wang et al. (2022). Wang et al. (2022) integrated organic vapor from the lowest volatility bin to $C^* \leq 10^{0.5}$ µg m⁻³ and regard them as condensable vapors. The mass flux of condensable vapors between gas and particle phase was calculated. Nie et al. (2022) calculated the contribution of condensing organic vapor to the formation of SOA. For organic vapor with relatively lower volatility ($C^* \leq 0.01$ µg m⁻³), the condensation to particle-phase was regarded as irreversible. We noticed that using the name "non-condensable organic vapors" and "condensable organic vapors" could lead to confusion, since "non-condensable organic vapors" can also reach the particle phase through gas-particle partitioning. Thus, we modified the classification, ELVOC and LVOC are classified as low volatility organic vapors ($C^* \leq 0.3$ µg m⁻³), while SVOC, IVOC and VOC fall into

another category regarded as high volatility organic vapors ($C^* > 0.3$ μg m⁻³). The corresponding sentences have been revised.

6. Line 308-313: I wonder if the authors can quantitatively estimate the contribution of the "non-condensable" organic vapors to the total organic aerosol mass to strengthen this point. The saturation vapor concentration for the gas-phase organic vapors have already been estimated. The organic aerosol mass loadings from SP-AMS are available. Then the particle-phase concentrations of these compounds can be calculated based on equilibrium partitioning and compared with the mass that FIGAERO is missing out (mass balance).

Reply: We appreciate the viewer for this valuable suggestion. We have estimated the contribution of high volatility organic vapors (SVOC+IVOC+VOC) to the OA concentration ($Estimated\ OA_{HVgas}$) based on the following equation:

$$Estimated\ OA_{HVgas} = \sum_i C_{i,g} f_i \tag{1}$$

[revised manuscript text omitted]